# Can Language Models Teach Weaker Agents? Teacher Explanations Improve Students via Personalization

**Swarnadeep Saha**    **Peter Hase**    **Mohit Bansal**
Department of Computer Science
University of North Carolina at Chapel Hill
{swarna, peter, mbansal}@cs.unc.edu

## Abstract

A hallmark property of explainable AI models is the ability to teach other agents, communicating knowledge of how to perform a task. While Large Language Models (LLMs) perform complex reasoning by generating explanations for their predictions, it is unclear whether they also make good teachers for weaker agents. To address this, we consider a student-teacher framework between two LLM agents and study *if, when, and how* the teacher should intervene with natural language explanations to improve the student's performance. Since communication is expensive, we define a budget such that the teacher only communicates explanations for a fraction of the data, after which the student should perform well on its own. We decompose the teaching problem along four axes: (1) *if* teacher's test time intervention improve student predictions, (2) *when* it is worth explaining a data point, (3) *how* the teacher should personalize explanations to better teach the student, and (4) if teacher explanations also improve student performance on *future* unexplained data. We first show that teacher LLMs can indeed intervene on student reasoning to improve their performance. Next, inspired by the Theory of Mind abilities of effective teachers, we propose building two few-shot mental models of the student. The first model defines an Intervention Function that simulates the utility of an intervention, allowing the teacher to intervene when this utility is the highest and improving student performance at lower budgets. The second model enables the teacher to personalize explanations for a particular student and outperform unpersonalized teachers. We also demonstrate that in multi-turn interactions, teacher explanations generalize and learning from explained data improves student performance on future unexplained data. Finally, we also verify that *misaligned* teachers can lower student performance to random chance by intentionally misleading them.[1]

## 1 Introduction

Teaching, or the ability to provide needed information in a way that is understood by others, is often considered an important property of Explainable AI [1]. When AI models "teach" by providing meaningful and interpretable explanations, it fosters transparency, warranted trust, and the ability for humans to make informed decisions based on AI recommendations. One way the goodness of an explanation can be judged is by its ability to communicate knowledge of how to solve a problem to other agents [2, 3]. Explanations fulfill this purpose not only by being informative but also by means of filling in specific gaps in the recipient agent's knowledge. This is enabled by the explainer having *theory of mind (ToM)*, understanding what the recipient does not know [2], and being able to *personalize* its explanations based on the recipient's needs. Recent work has argued that LLMs like GPT-3.5 now exhibit ToM, based on their ability to answer questions about mental states of

---

[1]Code for all experiments: https://github.com/swarnaHub/ExplanationIntervention

37th Conference on Neural Information Processing Systems (NeurIPS 2023).

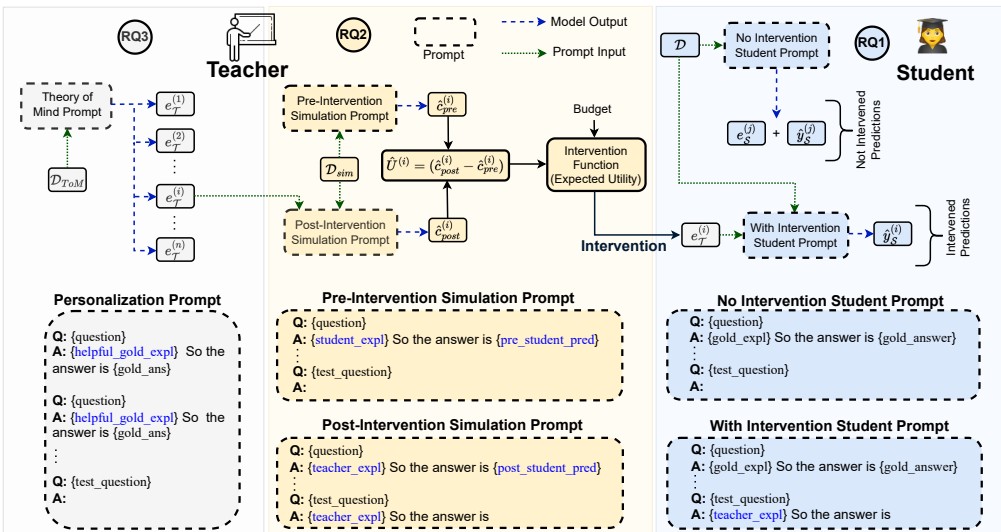

**Figure 1:** Overview of single-round of interaction between a teacher LLM and a student LLM, covering our first three research questions (with Fig. 6 showing RQ4, multi-round student-teacher interaction that builds on top of RQ1-RQ3). *RQ1:* The teacher randomly intervenes and communicates explanations to improve the student's performance (right part). *RQ2:* The teacher decides when to intervene by computing Expected Utility of Intervention using the Pre-Intervention and Post-Intervention Simulation prompts (middle part). *RQ3:* The teacher communicates personalized explanations that are more helpful for the student (left part).

hypothetical people in classical theory-of-mind tests [4].[2] However, we do not yet know how well language models can *teach* other agents to solve reasoning tasks via explanations.

In this work, we are motivated by this essential goal of evaluating explanations (specifically, Chain-of-Thought [7]) rationales) from the perspective of teaching and improving weaker agents in solving reasoning tasks. In order to improve smaller models' reasoning skills, recent works propose knowledge distillation by fine-tuning a smaller model on the reasoning steps generated by a larger model [8, 9, 10]. Yet, an important component of human teaching is understanding *when* and *how* the teacher should explain particular things to the student. Current distillation approaches do not evaluate a teacher's ability to identify when a student lacks understanding, and past work has not explored how to personalize teacher explanations to the student's needs. A smaller student model might already be good at answering certain questions but might require the teacher's intervention for some harder questions. When there are many things to teach the student and teaching is laborious, it is important to choose which problems merit explanation in order to improve teaching efficiency [11]. Moreover, for more effective teaching, it is desirable to have the teacher personalize its explanations to help a particular student, and a teacher that lacks understanding of the student's needs (i.e., lacks Theory of Mind) will be unable to do this [3].

Motivated by the efficiency of human explanations, we consider a student-teacher framework where a teacher model guides the reasoning of a student model, with the goal of improving the student's reasoning on current and future data. In order to do so, we explore a Theory of Mind-inspired approach, where the teacher simulates the student's behavior by building a *mental model* of the student. Our overall research question investigates whether the teacher's intervention (in the form of natural language explanations) can enable the student to make more accurate predictions both on explained as well as unexplained future data. However, communication is expensive, and therefore we assume that a cost is incurred each time the teacher intervenes with (communicates) an explanation to the student for a particular data point. We refer to this as the *intervention budget*, the percentage of test data points the teacher intervenes on. In order to comprehensively answer our overall research question, we further decompose the teaching problem into the following constituent questions:

1. **RQ1.** Can a teacher LLM intervene at test time to improve a student LLM's predictions?

---

[2]There have also been strong arguments *against* the presence of ToM in current LLMs [5, 6]. In this paper, we do not thoroughly test for ToM abilities in LLMs, but instead we focus on measuring teaching performance in terms of a teacher's positive effect on student performance.

2. **RQ2.** Given a fixed intervention budget, when should the teacher intervene (i.e., on which data points), in order to maximize student performance?
3. **RQ3.** Given a set of intervention data, can a teacher model personalize its explanations for a student model to improve student performance?
4. **RQ4.** In multi-turn interactions, do teacher explanations generalize and improve student performance across data points (beyond the explained samples)?
5. **RQ5.** Can *misaligned* teacher LLMs lower student performance by providing misleading explanations to the student?

We answer RQ1 by assuming that the teacher intervenes on random data points in four different settings: using a human or LLM teacher, and, when the teacher is an LLM, using an LLM student that is either weaker or stronger than its teacher (§5.1). Across three different reasoning tasks (StrategyQA, GSM8k, and CommonsenseQA) and two different model families (Flan-T5 and LLaMA), we observe that (1) teacher LLMs can effectively intervene on student reasoning, improving student performance on the end task, and (2) more intervention typically leads to a monotonic increase in student performance, though model teachers are not as good as human teachers. Fig. 1 shows the intervention process and the two student prompts (in the right part of Fig. 1) that are used to generate predictions.

RQ2 explores how to intelligently select which data points to explain for the student model, in order to improve teaching efficiency (§5.2). Past work in cognitive science also considers teaching efficiency in young children by deciding what to teach by maximizing the learner's expected utility of learning [12]. With a similar motivation, we develop an Intervention Function that is inspired from the principle of a teacher having a Theory of Mind. In particular, the teacher builds a *mental model* of the student's reasoning process, with the goal of intervening only on samples that are most likely to maximize student performance. Our Intervention Function is based on Expected Utility, in which the teacher first estimates the utility of an intervention by simulating the student's prediction pre-intervention (without intervention) and post-intervention (with intervention), then constructs a rank ordering of the samples according to this utility (see the middle part of Fig. 1). The teacher builds this mental model in a few-shot manner, only assuming access to the student's predictions pre- and post-intervention for a few samples. We demonstrate that our proposed Intervention Function based on Expected Utility (1) outperforms other baseline Intervention Functions, (2) improves student performance when the teacher is not 100% accurate, and (3) enables weaker LLMs to teach stronger ones, unlike with random intervention in RQ1.

Next, in RQ3, we explore *how* the teacher should explain data points to a particular student model, including how the teacher can personalize explanations for a student model (§5.3). That is, after deciding which data points to intervene on (RQ2), we decide how the teacher should explain those data points. A clear limitation of the teacher just generating explanations as if it is solving the task is that the explanations are not at all personalized for the student. Given that good explanations are designed to fill in gaps in student knowledge [2], we believe that equipping the teacher with basic personalization skills will improve its teaching ability. With this motivation, we propose another few-shot mental model for the teacher that encourages it to tailor its explanations to be helpful for the particular student model it is teaching. The teacher builds this model by conditioning on a few demonstrations of 'useful' human explanations that rectify a student's answer, thereby encouraging explanations that are more likely to help the student (see Fig 1 for an example of the teacher's personalization prompt). We demonstrate this prompt's effectiveness against unpersonalized explanations that are generated by prompting the teacher with random human explanations, showing that LLMs can personalize their explanations.

RQ4 tests whether LLMs can teach student models to generalize to new unexplained examples (§5.4), rather than improve their reasoning at test-time (RQ1-RQ3). In other words, we now explore the ability of LLMs to teach using the teaching components introduced in RQ2 and RQ3 of when and how to explain samples. This leads us to explore a *multi-round* interactive setting, where each round consists of the teacher selecting a set of best points to explain (according to RQ2) and generating explanations for them (according to RQ3). The student then conditions on these *teacher explanations* as in-context demonstrations to perform the reasoning task on future unexplained samples. We demonstrate that teacher explanations indeed generalize and improve student performance on unexplained data.

Finally, in RQ5, we investigate the negative implications of teacher explanations on student LLMs (§5.5). Given that LLMs can improve student agents, we also want to test whether they can lower student performance. If a *misaligned* teacher provides non-factual explanations in scenarios where the

student directly adopts them, does that lead to a drop in student performance? In fact, we show that teacher models can lower student performance to random chance by intervening on data points with the intent of misleading the student. This has potential implications for LLMs giving explanations in a context where other agents adopt them with unwarranted trust in their correctness.

In summary, our comprehensive studies highlight the ability of LLMs to teach and improve weaker LLMs, demonstrated via improvements on explained test examples as well as future unexplained data. Broadly, equipping LLMs with an ability to effectively and efficiently teach, opens the door to (1) using LLMs as personalized tutors for humans (where efficiency is critical), (2) distilling knowledge into weaker or more compute-efficient student models, and (3) improving human decision making via AI recommendations and explanations.

## 2   Related Work

**Evaluating Explanations in Teacher-Student Games.** Several past works evaluate explanations in the context of a student-teacher communication game [13, 14, 15, 16, 17]. The teacher communicates explanations to the student with one of two objectives: (1) evaluating whether explanations help students to simulate the teacher better, or (2) whether explanations can directly teach students to better perform a task. *Simulatability*, or the student's ability to simulate the teacher's own answer, is seen as a measure of the explanation's faithfulness, rather than a direct measure of whether explanations help students learn the task itself [18, 19]. Our work is focused on the second research goal of evaluating explanations from the perspective of teaching weaker agents. Prior work has shown that human explanations can teach LLMs [7, 20] and LLMs can also teach themselves from their own explanations or feedback [21, 22]. But it remains to be shown whether LLMs can also teach weaker agents. A few recent works also share a similar goal as ours and they distill knowledge [23] directly into the student model by finetuning it on the explanations from the teacher model [8, 9, 10, 24]. However, these distillation methods do not consider the important aspects of communication cost between two agents, its trade-off with student performance, and how the teacher may build mental models of the student to decide *when* and *how* to communicate explanations to the student. Recent studies have also evaluated explanations in the context of human-AI collaboration, for their plausibility [25, 26], usefulness to human learning [27, 28], and for improving human-AI team performance [29, 30]. Different from these, we analyze model-model interactions, with the goal of understanding how effectively LLMs can teach weaker systems to solve a task.

**Theory of Mind in AI.** A body of work demonstrates that humans regularly infer and make decisions based on the mental states of other agents, also known as Theory of Mind (ToM) [31, 3, 2, 32]. This has motivated works on computational language acquisition using ToM [33, 34, 35]. There have been recent works arguing both for and against the presence of Theory of Mind in Large Language Models [36, 4, 6, 37]. Theory of Mind has been successfully applied to improve human-AI collaboration in robotics [38, 39, 11]. In this work, we design prompts that are motivated by a teacher having a Theory of Mind to efficiently intervene and improve a student's reasoning capabilities.

## 3   Problem Setup

**Student and Teacher.** We assume a two-agent communication game between a student $\mathcal{S}$ and a teacher $\mathcal{T}$, where the goal is to teach the student to solve a particular task interactively. Here, we use an LLM as the student. To explore a range of student and teacher capabilities, we consider both human and model teachers, while typically using a student model that is measurably weaker than its teacher. Following past work, an LLM with more parameters is considered a stronger model due to its better performance across a range of tasks (including the ones we consider in our studies). In the scope of our study, when the teacher is not a human, both $\mathcal{S}$ and $\mathcal{T}$ are LLMs, prompted with a set of demonstrations of the task $\mathcal{D}$ (typically, 4-8 examples). Each demonstration $d^{(i)} \in \mathcal{D}$ is a triple $(x^{(i)}, y^{(i)}, e_{\mathcal{H}}^{(i)})$ consisting of an input $x^{(i)}$, the output $y^{(i)}$, and a human-written explanation $e_{\mathcal{H}}^{(i)}$ that answers the question of *why* the data point has the output it has [1]. In our tasks, the explanation may include background knowledge or intermediate reasoning steps that are helpful for obtaining the answer. By organizing $\mathcal{D}$ into Chain-of-Thought prompts, both $\mathcal{S}$ and $\mathcal{T}$ are equipped with the ability to generate explanations and predict labels for new samples.

**Single-Round Intervention.** The first problem setup we consider involves the teacher deciding whether or not to intervene for a single test problem. In this setting, the student's goal is to answer

the problem correctly, and the teacher can choose to intervene for individual problems to assist the student. Thus, given a test data point $t^{(i)}$, we have the following two scenarios:

- **No Intervention:** When the teacher chooses not to intervene, the student generates both the explanation $e_{\mathcal{S}}^{(i)}$ and the prediction $\hat{y}_{\mathcal{S}}^{(i)}$, by conditioning on the $\mathcal{D}$ task demonstrations and the test input $t^{(i)}$. This is done using Chain-of-Thought prompting [7].

- **Intervention:** When the teacher does choose to intervene, it communicates its generated explanation to the student. Here the student only generates a prediction $\hat{y}_{\mathcal{S}}^{(i)}$ by conditioning on the $\mathcal{D}$ task demonstrations, the test input $t^{(i)}$, and the corresponding teacher explanation $e_{\mathcal{T}}^{(i)}$. For the tasks and datasets we consider, explanations provide helpful background information or reasoning steps but do not directly reveal the test label, so it never directly gives away the answer.

Fig. 1 shows the 'No Intervention' and 'With Intervention' student prompts and the overall intervention process. Note that in terms of the prompts used for both these scenarios, the only difference is in the source of explanation (student vs teacher) for the test point. When the teacher is a human, intervention happens with a human-written explanation (crowdsourced in the datasets we rely on).

**Communication Cost.** In Single-Round Intervention, the teacher could maximize student performance by simply always intervening on the student's reasoning. We bring in a natural constraint from Rational Speech Acts theory, i.e., communication is costly and should only be undertaken if it furthers a communicative goal [40]. Hence, we assume that a communication cost is incurred each time the teacher intervenes with an explanation to the student. We also note that this cost is only with respect to an agent's communication (and assume that the teacher can always generate explanations for itself). Unless otherwise stated, communication happens one-way from the teacher to the student in the form of explanations. We set a limit to the number of points that the teacher can intervene on, referred to as the *intervention budget*, and we assume the cost to be uniform for all data points. Across all our experiments, we vary the intervention budget between $\{0\%, 20\%, 40\%, 60\%, 80\%, 100\%\}$. A budget of 0% means the student generates its own explanation as it predicts each data point, while a budget of 100% means the student leverages the teacher's explanation for every data point. Later, in Sec. 5.2, we introduce the teacher *Intervention Function*, which the teacher uses to decide which points to intervene on given its fixed intervention budget.

**Multi-round Intervention.** Here, the goal of the teacher is to provide explanations to the student that help it generalize across samples, rather than leading the student to the correct answer only for the explained data points. Thus, we allow the teacher to explain data points that are then added to the student model's prompt, but we forbid the teacher from intervening on future test points. If the teacher can improve the quality of the student model's prompt, student performance on the test data should improve. In our experiments, this process occurs in a few steps: (1) the teacher picks points to explain, (2) the teacher explains those points, (3) the points are added to the student prompt with the teacher's explanations, and then (4) the student predicts labels for the remaining test data. Further implementation details are given in Sec. 5.4.

## 4 Experiment Setup

We consider complex natural language reasoning tasks, motivated by two specific needs. First, the task should be hard enough for current LLMs that explanations can provide useful knowledge to the models. Second, it allows us to study free-text rationales, which are more flexible than input attribution methods [41, 42, 43] and benefit many reasoning tasks [7, 44]. We experiment with three reasoning tasks: (1) StrategyQA [45], (2) GSM8k [46], and (3) CommonsenseQA [47] (details in Appendix B). Both StrategyQA and CommonsenseQA contain explanations in the form of relevant facts, thus requiring the student to reason over them to produce an answer. We also manually verify (up to 100 samples) that the explanations do not explicitly leak the answer. For GSM8k, since the reasoning steps explicitly derive the answer, providing the entire explanation during intervention will lead to answer leakage [17]. Hence, the teacher communicates a partial explanation (specifically, only the first step of the rationale), allowing the student to leverage it as a hint to derive the final answer. We experiment with two state-of-the-art open-source LLMs of varying sizes, ranging from 780M to 65B parameters. Specifically, we use two encoder-decoder and decoder-only models as student and teacher: (1) Flan-T5-Large and Flan-T5-XL [48], and (2) LLaMA-7B, LLaMA-13B, and LLaMA-65B [49]. Refer to Appendix A for more details of student and teacher models.

# 5 Experiment Results

## 5.1 RQ1: Can a teacher LLM intervene at test time to improve a student LLM's predictions?

Our first research question asks if LLMs can improve students by intervening on their reasoning at test time. While the main goal is to analyze the behavior of model-based teachers, we also experiment with human teachers to establish a ceiling on the capabilities of an LLM teacher. These human teachers are people who authored the (human) explanations in the datasets we experiment with and were crowdsourced in prior works.

**Study Design.** We compare the accuracy obtained by the student model at different intervention budgets. For the purpose of this study, the intervention happens at random data points while we vary the student and teacher. In particular, we compare four intervention setups: (1) a human teacher paired with a smaller student model, (2) a human teacher paired with a larger student model, (3) a larger teacher model paired with a smaller student model, and (4) a smaller teacher model paired with a larger student model. For the main experiments, the student and the teacher are chosen from the same model family.

**Main Results.** Figure 2 shows the results on StrategyQA with Flan-T5 models. A human teacher's intervention on the explanations of both smaller and larger Flan-T5 models exhibits a monotonically increasing accuracy trend. Larger model teachers can also improve smaller student models. Flan-T5-Large obtains an accuracy of 58% when always utilizing its own explanations but obtains up to 63% accuracy when reasoning with the larger Flan-T5-XL's explanations. Intuitively, a larger student model does not benefit from a smaller teacher model's explanations, as we observe a monotonically decreasing trend. Our results generalize to other models (LLaMA), datasets (CommonsenseQA, GSM8k) and even when the student and the teacher belong to different model families. In fact, when the teacher (LLaMA-65B) is much stronger than the student (LLaMA-7B), the margin of improvement is also

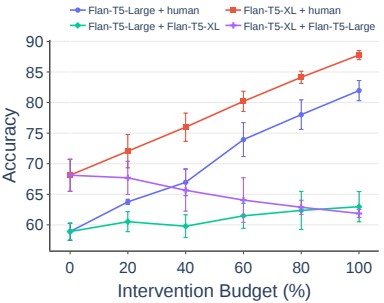

**Figure 2:** RQ1: Comparison of random intervention by different Teacher Models on different Flan-T5 Student Models at different intervention budgets for StrategyQA. A + B = A student, B teacher.

higher, about 8% (statistically significant with $p = 0.01$). See Appendix D for these additional RQ1 results. In summary, we conclude that: *for complex reasoning,LLMs can indeed effectively intervene and improve weaker models, and more intervention typically leads to better performance, although humans explanations improve more.*

## 5.2 RQ2: Given a fixed intervention budget, when should the teacher intervene (i.e., on which data points), in order to maximize student performance?

So far, we have demonstrated that random teacher intervention benefits student models. But a good teacher does not randomly pick problems to help a student with. Each intervention also has an associated communication cost and hence, it is desirable to be able to improve student performance while reducing the cost. In this research question, we investigate better strategies for choosing data points to intervene on. We call these strategies *Intervention Functions* that produce a rank ordering of the samples, and, given a fixed budget, the teacher intervenes on the highest-ranked samples.

An intervention is useful if the student's confidence in the gold answer increases with intervention (i.e., with teacher's explanation) compared to without it (i.e., with its own explanation). Here confidence is simply the likelihood that the model assigns to the correct answer i.e., we take the logits from the last layer of the model and normalize them to get the correct answer's probability. Computing expected utility, however, depends on two quantities: (1) the student's *true confidence* measures with and without intervention, and (2) the *gold answers* against which the confidence is computed. It also incurs a two-way communication cost, one for the teacher to communicate its explanation to the student and another for the student to communicate back its confidence to the teacher. Thus, we propose an Intervention Function based on the *Expected Utility* of intervention, which relies on estimates of student confidence, and we consider two setups depending on whether the teacher knows the gold label. Ideally, a teacher is expected to be an expert in the concerned task (e.g., if the teacher

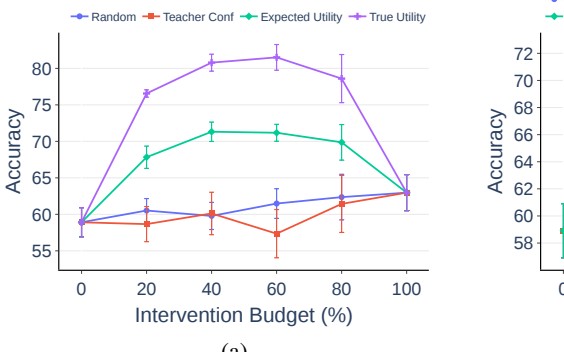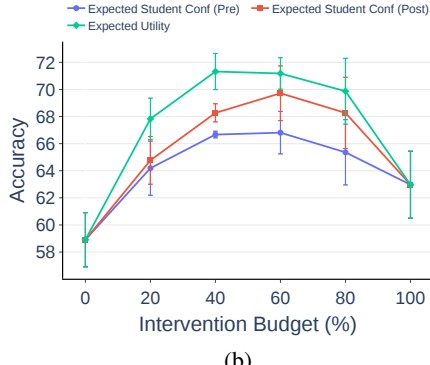

| (a) | (b) |

**Figure 3:** RQ2: (a) Comparison of different Intervention Functions on StrategyQA with a smaller student (Flan-T5-Large) and a larger teacher (Flan-T5-XL). (b) Ablation of Expected Utility.

is a human or a powerful model that obtains high accuracy). When the teacher does not have access to gold answers, we treat the teacher's answers as gold answers when computing Expected Utility.

**Expected Utility Intervention Function.** The teacher computes the *Expected Utility* of intervention by simulating the student's predictions using a *mental model* of the student. In order to build this mental model, we assume that the teacher has observed the student on a few samples and has access to $d$ demonstrations $\mathcal{D}_{sim}$ of the student's predictions with and without intervention, denoted as:

$$\mathcal{D}_{sim} = \{x^{(i)}, y^{(i)}, e_{\mathcal{S}}^{(i)}, e_{\mathcal{T}}^{(i)}, \hat{y}_{pre}^{(i)}, \hat{y}_{post}^{(i)}\}_{i=1}^{d}$$

where $x^{(i)}$ and $y^{(i)}$ are the input and output respectively; $e_{\mathcal{T}}^{(i)}$ and $e_{\mathcal{S}}^{(i)}$ denote the student and teacher explanations respectively; and $\hat{y}_{pre}^{(i)}$ and $\hat{y}_{post}^{(i)}$ refer to the student predictions with student explanation (pre-intervention) and teacher explanation (post-intervention) respectively. Using these demonstrations, the teacher builds a few-shot mental model of the student and predicts two quantities for a given test question – (1) **Pre-intervention Expected Student Confidence** ($\hat{c}_{pre}$): The teacher conditions on the pre-intervention demonstrations $\mathcal{D}_{sim}^{pre} = \{x^{(i)}, y^{(i)}, e_{\mathcal{S}}^{(i)}, \hat{y}_{pre}^{(i)}\}_{i=1}^{d}$ to simulate the student's confidence on the gold answer, had it been using its own (student) explanation, and (2) **Post-intervention Expected Student Confidence** ($\hat{c}_{post}$): The teacher conditions on the post-intervention demonstrations $\mathcal{D}_{sim}^{post} = \{x^{(i)}, y^{(i)}, e_{\mathcal{T}}^{(i)}, \hat{y}_{post}^{(i)}\}_{i=1}^{d}$ to estimate what the student's confidence would be if it had used the teacher's explanation. The teacher computes these confidence estimates as if it were the student (refer to Fig. 1 for the prompts), essentially learning to simulate the student by conditioning on the appropriate demonstrations and then generating an answer to the question. Then the Expected Utility $\hat{U} = (\hat{c}_{post} - \hat{c}_{pre})$ is given by the difference between the two confidence measures. The teacher finally constructs a rank ordering of the test data points based on this expected utility. This utility-based ordering encourages the teacher to pick points where it thinks the student will answer correctly with intervention but incorrectly without.

**Other Intervention Functions.** To analyze how well our proposed Intervention Function performs, we compare it with the following Intervention Functions. Our first baseline is the **Random Intervention Function** from RQ1. Next, we compare with an Intervention Function that ranks the samples based on the **Teacher Confidence** – when the teacher is most confident about a question, it intervenes. Our next two baselines are ablations of Expected Utility: (a) **Pre-Intervention Expected Student Confidence** – We rank samples based on the expected student confidence with no intervention (i.e., lower this confidence, the higher the likelihood of useful interventions), and (b) **Post-Intervention Expected Student Confidence**: We rank samples based on the expected student confidence with intervention (i.e., higher this confidence, higher is the likelihood of useful interventions. Finally, as upper bounds of Intervention Functions, we assume that the student communicates its *true* confidence values to the teacher (which for post-intervention, incurs a both-way communication cost of the teacher sending its explanation, followed by receiving the student's confidence). Using the true confidence measures, we compute **True Utility**.

**Main Results: How does Expected Utility compare to True Utility?** Figure 3 compares different Intervention Functions with Flan-T5-XL as the teacher and Flan-T5-Large as the student on StrategyQA. Across different methods, we analyze accuracy obtained at lower communication costs (e.g.,

20%) as well as highest accuracy obtained, independent of any budget constraints. Our primary observation from Figure 3(a) is that Expected Utility improves student accuracy by up to 7 points at a low communication cost of 20%. Expected Utility also peaks at an accuracy of 71% with only 40% intervention. Since model-based teachers are not always perfect, increased intervention beyond 60% leads to a drop in student accuracy (e.g., in the last 20% of the intervention, the student accuracy drops from $69\% \rightarrow 63\%$). When the student communicates its confidence scores to the teacher, the teacher is able to compute the true utility of intervention, which unsurprisingly, leads to a much higher accuracy of 76% at 20% cost and an overall high of 81% accuracy. Nevertheless, estimating expected utility is cheaper and our results also suggest that a better mental model could further improve performance. Ranking by teacher confidence is ineffective because it is not an indicator of the student's capabilities. Next, in Figure 3(b), we show that ranking by utility outperforms ranking by either pre or post-intervention confidence scores. In summary, *with access to only a few demonstrations of student behavior, a teacher can build an effective mental model of the student and intervene such that the student obtains a much higher accuracy at low communication costs.*

**When the teacher does not have access to gold answers, can we compute Expected Utility with respect to teacher answers?** Teachers can be inaccurate and may not even have access to gold answers. In such scenarios, can we treat the teacher as the gold standard and compute utility with respect to the *teacher's answers*? We explore this in Figure 4, comparing Expected Utility to Random Intervention and Student Least Confidence. The latter denotes that when the student is least confident about any of the answer options, it is more likely to answer incorrectly and hence will benefit from intervention. We observe that Expected Utility, computed with teacher answers, also leads to up to 2 points improvement in accuracy at 20% budget, which is also within 1% of the accuracy (63.60%) obtained with 100% communication cost. In Appendix Table 9, we conduct the same experiment with a much stronger teacher

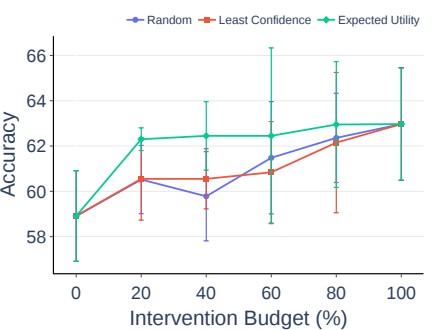

**Figure 4:** RQ2: Comparison of different Intervention Functions on StrategyQA with a Flan-T5-Large student and a Flan-T5-XL teacher, with no access to gold labels.

(LLaMA-65B) and a weaker student (LLaMA-7B) and obtain even stronger evidence of this result. Stronger teachers like LLaMA-65b are significantly better at solving reasoning tasks and thus their predicted labels will mostly match the gold labels. Hence, even if we rely on the teacher's predictions for computing expected utility, it improves student accuracy by up to 5 points (statistically significant with $p = 0.02$), further closing the gap between 'with and without gold label' scenarios. In summary, we conclude that *imperfect teacher LLMs can also successfully intervene by building mental models of students that do not rely on ground-truth answers.* Appendix E contains additional results for RQ2 with other models and datasets.

### 5.3 RQ3: Given a set of intervention data, can a teacher model personalize its explanations for a student model to improve student performance?

The previous RQ showed how the teacher may build a few-shot mental model of the student to decide *when* to intervene, given a fixed budget. Upon intervention, the teacher communicates an explanation generated by prompting the model with random human explanations. This leads to an unpersonalized teacher that assumes that the explanation it generates in order to solve the task will be automatically helpful for the student. However, an effective teacher should tailor its explanations to fill in gaps in the student's knowledge [2]. With this motivation, the teacher builds another few-shot mental model of the student, this time generating helpful explanations that are more likely to benefit the student.

**Teacher's Explanation Personalization** Prompt. Helpful human explanations are those that rectify a student's answer i.e., cause the student's answer to flip from incorrect (when using its own explanation) to correct (when using human explanation). We assume that the teacher has observed the student on $d$ demonstrations $\mathcal{D}_P$ of exclusively helpful human explanations, denoted as: $\mathcal{D}_P = \{x^{(i)}, y^{(i)}, e_{\mathcal{H}}^{(i)}, e_{\mathcal{S}}^{(i)}\}_{i=1}^d$ where $e_{\mathcal{H}}^{(i)}$ and $e_{\mathcal{S}}^{(i)}$ denote (helpful) human and (not helpful) student explanations respectively. The teacher conditions on these demonstrations to generate explanations

for the student. Fig. 1 shows an example of a personalization prompt. With such a prompt, teacher explanations are steered toward only those explanations that help the student.

**Baselines.** We compare personalized teachers with unpersonalized ones that condition on *random* human explanations. Appendix F also reports results with unpersonalized rationales, that are post-hoc explanations ('The answer is X because Y') and not Chain-of-Thought ('Y. So the answer is X').

**Main Results.** Fig. 5 shows the results on StrategyQA with Flan-T5-Large as the student and Flan-T5-XL as the teacher. Both unpersonalized and personalized teachers choose intervention samples based on Expected Utility, as defined in RQ2. We observe that a personalized teacher improves the accuracy further both at lower budgets (by 2% at 20% cost) but also overall, obtaining a peak accuracy of 72.63%. However, unlike the strong supporting evidences we obtain for RQ1 and RQ2, the effect of personalization is comparatively weaker. Hence, we further test this research question with a LLaMA-65B teacher and a LLaMA-7B student in Appendix F. While scaling up the teacher model points to stronger evidence of personalization (e.g., 2.4% better student accuracy), the results are still not statistically significant ($p = 0.09$). Hence, we conclude that: *personalizing teacher explanations can further benefit the students, although our results currently suggest that the effect size may be small*. We hope that future work is further able to explore explanation personalization with even stronger teacher models like GPT-4. In Appendix F, we show some comparative instances of unpersonalized and personalized explanations.

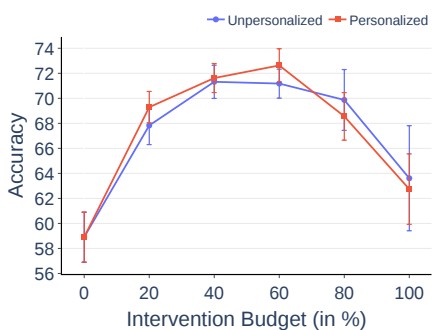

**Figure 5:** RQ3: Comparison of unpersonalized and personalized teacher (Flan-T5-XL) explanations on student (Flan-T5-Large) accuracy for StrategyQA.

### 5.4 RQ4: In multi-turn interactions, do teacher explanations generalize and improve student performance across data points (beyond the explained samples)?

In the previous RQs, we showed that teacher explanations improve student predictions for the samples that the teacher explains. RQ4 explores whether teacher explanations also generalize to new instances that the teacher has not explained. In other words, this studies if the student can perform Chain-of-Thought reasoning by only conditioning on teacher LLM explanations rather than human's.

**Study Design.** We consider a multi-turn teaching setup (Fig. 6), in which at each turn the teacher chooses to explain a few samples from a pool of unexplained examples which are then added to the student's prompt. The prompt consists of demonstrations of the teacher's explanations and predictions. The student then conditions only on these in-context examples (without any human demonstrations) to generate predictions for the test samples (where there is no teacher intervention). For choosing the data points to explain at each round, we use the Expected Utility Intervention Function (from RQ2), and for generating the teacher

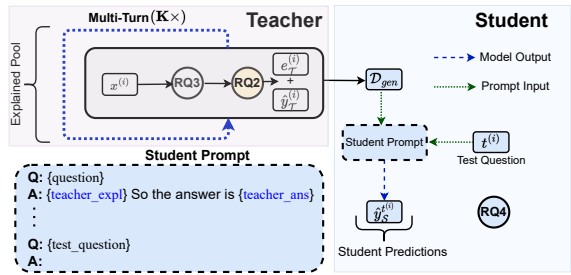

**Figure 6:** Overview of multi-turn student-teacher interaction showing *RQ4*. At each turn, the teacher chooses some samples to explain and the student conditions on them to make predictions on future unexplained data.

explanations, we leverage the ToM prompt (from RQ3). We say that teacher explanations generalize if conditioning on demonstrations of explained points improves upon demonstrations with no explanations (i.e., only QA pairs) or self-explanations (i.e., demonstrations with student explanations and predictions). We consider five rounds in total with LLaMA-7B as the student and LLaMA-65B as the teacher, adding two explained samples in each round. We compare the student accuracy after each round with teacher-explained, student-explained, and unexplained demonstrations.

**Main Results.** Fig 7 shows the results. We observe that teacher explanations improve student performance on future unexplained test points as well by a significant 6 points ($55\% \rightarrow 61.6\%$).

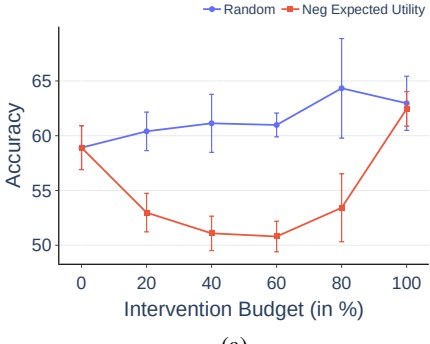
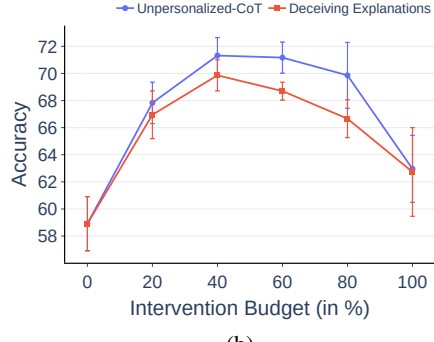

(a)                (b)

**Figure 8:** RQ5: (a) Negative Implication of RQ2: Comparison of intervention based on negative expected utility with random intervention on StrategyQA. (b) Negative Implication of RQ3: Comparison of an unpersonalized teacher (generating explanations conditioned on random human explanations) versus a deceiving teacher (generating explanations conditioned on wrong explanations).

While student accuracy with unexplained samples improves after adding more demonstrations (55% → 59%), accuracy with teacher-explained demonstrations is better by up to 4 points ($p = 0.04$) after each round. Interestingly, when the student conditions on self-explained demonstrations, its performance decreases with increasing rounds. We hypothesize that this is because the student might be overfitting to its own worse explanations, leading to bad predictions. In summary, we conclude that: *teacher LLMs can teach student models to perform well on their own when given new test data.* LLMs with even longer context windows will allow adding more explained samples in each round and repeating the teaching process for more rounds. We hope that our initial promising results will encourage more exploration in multi-round teaching with LLMs.

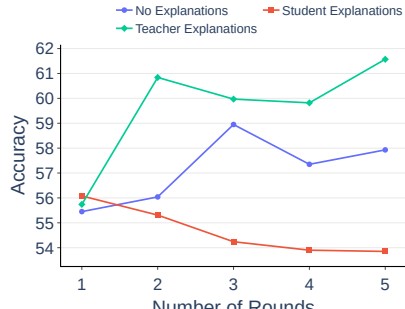

**Figure 7:** RQ4: Multi-round student-teacher interaction comparing student accuracy on unexplained test points with unexplained, student-explained and teacher-explained demonstrations.

### 5.5 RQ5: Can *misaligned* teacher LLMs lower student performance by providing misleading explanations to the student?

If teacher LLMs can successfully build mental models of student LLMs, a natural follow-up question regards whether communicating misleading explanations can also weaken student models. Here we briefly describe our study design and findings, with more details in Appendix H. First, the teacher intervenes in *increasing* order of expected utility, prioritizing samples where the utility is lowest. Second, we make the teacher condition on *incorrect* answers and *non-factual* human explanations that we manually generate by perturbing (correct) human explanations. We show the results in Fig. 8(a) and Fig. 8(b). Ranking data points by negative expected utility allows the teacher to reduce student accuracy to random chance at 60% intervention. Next, Fig. 8(b) illustrates that the teacher can condition on non-factual explanations to successfully generate worse explanations that reduce the student accuracy by 2 points, relative to the improvement of benign explanations. Thus, we conclude that *teacher LLM explanations can be influential for student LLMs in both good and bad ways.*

## 6 Conclusion

We demonstrated that LLMs can teach weaker student models to improve their performance on reasoning tasks for both explained and unexplained future data. The teacher builds two few-shot mental models of the student, one predicting which data points to intervene on and another generating personalized explanations that the student can efficiently learn from.

**Limitations & Broader Impacts.** See Appendix for limitations and broader impacts discussion.

## Acknowledgments

We thank Shiyue Zhang and Archiki Prasad for useful feedback, as well as Stephen Casper and Miles Turpin for suggestions regarding experiments. This work was supported by NSF-CAREER Award 1846185, NSF-AI Engage Institute DRL-2112635, DARPA MCS Grant N66001-19-2-4031, and Google PhD Fellowships. The views contained in this article are those of the authors and not of the funding agency.

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

## Broader Impacts

We hope that our findings can help improve the understanding and evaluation of Chain-of-Thought rationales, in order to better understand the behavior of LLMs and make them more interpretable. Through the first four research questions, we demonstrate that teacher LLMs can successfully build mental models of weaker agents to improve their performance. Modern LLMs like GPT-4 may generate non-factual explanations [50] that have the potential to inadvertently harm weaker agents, especially in a context where other agents adopt them with unwarranted trust in their correctness. We verify this through our final research question. In general, we do not foresee specific ethical risks arising from this work that do not already apply to the general use of Large Language Models, such as the potential to generate harmful or toxic content [51].

## Limitations

While teacher LLMs generate better explanations via personalization, the human explanations are unpersonalized i.e., collected without any particular student in mind. In spite of that, we observe that intervention with human explanations proves to be helpful in most cases. It remains to be seen whether human explanations that are directed toward improving a particular student model can lead to further improvements. Next, we make a simplifying assumption that the communication cost is uniform across all samples. Non-uniform costs (e.g., measured based on the number of tokens or reasoning steps) such that longer explanations incur larger costs is an interesting direction for future work. We also note that while both student and teacher generate explanations with the goal of improving student predictions, the predictions may still be unfaithful to the reasoning steps.

## A  Student and Teacher Models

We experiment with two state-of-the-art open-source LLMs of varying sizes, ranging from 780M to 65B parameters. Specifically, we use two encoder-decoder and decoder-only models as student and teacher: (1) Flan-T5-Large and Flan-T5-XL [48], and (2) LLaMA-7B, LLaMA-13B, and LLaMA-65B [49]. Typically, the student model is assumed to be smaller than the teacher model. But some experiments will also involve smaller teachers and larger students. All models generate text using greedy decoding, prompted with either 4-8 demonstrations. Unless otherwise stated, the demonstrations are randomly chosen from the training samples. For StrategyQA, we report results on the validation split, while for CommonsenseQA and GSM8k, our experiments are on the test split. To account for variance, we conduct experiments with at least three different seeds. We report accuracy for all tasks, and error bars in plots and tables represent the standard deviation across seeds.

## B  Datasets and Prompts

We experiment with the following three reasoning datasets: (1) StrategyQA [45], a set of open-domain questions where the required reasoning steps are implicit in the question, (2) GSM8k [46], which includes multi-step math reasoning problems, and (3) CommonsenseQA [47], a multiple-choice QA task focusing on commonsense reasoning. We use the reasoning steps in StrategyQA and GSM8k as the multi-step rationales, and for CommonsenseQA, we rely on the ECQA dataset [52], which is annotated with commonsense facts supporting the correct option and refuting the incorrect options. All datasets are licensed under the MIT license. Fig. 9 shows the student prompts for the three tasks of StrategyQA, CommonsenseQA, and GSM8k. Fig. 10 shows the pre- and post-intervention student simulation prompts for the teacher model.

## C  Compute and Reproducibility

We conduct experiments either on A100 Google Cloud instances or on internal A6000 GPU servers. The LLMs (Flan-T5 and LLaMA) and the datasets used in our studies are publicly available. For reproducibility, we are making our code available as part of the supplementary material.

| StrategyQA |
|---|
| **Q:** Are more people today related to Genghis Khan than Julius Caesar?
**A:** Julius Caesar had three children. Genghis Khan had sixteen children. Modern geneticists have determined that out of every 200 men today has DNA that can be traced to Genghis Khan. So the answer is yes

**Q:** {test_question}
**A:** |

| CommonsenseQA |
|---|
| **Q:** What might a person see at the scene of a brutal killing?
**Answer Choices:**
**Choice 1:** bloody mess
**Choice 2:** pleasure
**Choice 3:** being imprisoned
**Choice 4:** feeling of guilt
**Choice 5:** cake
**A:** Bloody mess is covered or stained with blood. A person might see a bloody mess at the scene of a brutal killing. Pleasure is about what a person sees at the scene of a brutal killing and one cannot be happy to see such brutality. You can't see someone in jail at the brutal killing scene. Feeling of guilt doesn't come as the killing is brutal or merciless. Cake is baseless and weird to think as it is a brutal killing scene and not a bakery. So the correct choice is 1

**Q:** {test_question}
**Answer Choices:**
**Choice 1:** {option_1}
**Choice 2:** {option_2}
**Choice 3:** {option_3}
**Choice 4:** {option_4}
**Choice 5:** {option_5}
**A:** |

| GSM8k |
|---|
| **Q:** Natalia sold clips to 48 of her friends in April, and then she sold half as many clips in May. How many clips did Natalia sell altogether in April and May?
**A:** Natalia sold 48/2 = 24 clips in May. Natalia sold 48+24 = 72 clips altogether in April and May. So the answer is 72

**Q:** {test_question}
**A:** |

**Figure 9:** Examples of student prompts for different tasks with one demonstration.

## D   RQ1: Additional Results

**Results with Flan and LLaMA Models.**   In Table 1, we report the accuracy obtained by different students and teachers (based on Flan-T5 models) on the StrategyQA task. We draw similar conclusions as Flan-T5 with other LLMs, specifically LLaMA-7B and LLaMA-65B models on the StrategyQA dataset (Table 2). In fact, when the teacher is stronger like a LLaMA-65B, the margin of improvement is also higher, about 8%. The overall trends also align – increasing for weaker students and decreasing for stronger students.

**Results on other Datasets.**   Our conclusions generalize across datasets too. Table 3 presents the results on CommonsenseQA with Flan-T5 models. CommonsenseQA is an easier task and Flan-T5 models obtain accuracies of 85% and 92% when generating their own explanations. While Flan-T5-Large still benefits from human explanations, the larger model does not, perhaps because it already starts at a high 92% accuracy. Finally, in Table 4, we present the results on GSM8k with LLaMA models. Note that in GSM8k, a student has access to partial explanations from the teacher, but even then we observe that these prove to be useful prompts for the student to complete their chain-of-thought, leading to up to 8-9% increase in accuracy with human teachers and 3% with model teachers.

| Pre-Intervention Student Simulation |
| --- |

Simulate an AI model's answer for the given question.

**Q:** Will the Albany in Georgia reach a hundred thousand occupants before the one in New York?
**AI Predicted Answer:** Albany, Georgia is a city in the U.S. state of Georgia. Albany, Georgia has a population of 59,080. The population of New York is 365,040. So the answer is no

**Q:** {question}
**AI Predicted Answer:**

| Post-Intervention Student Simulation |
| --- |

Simulate an AI model's answer for the given question.

**Q:** Will the Albany in Georgia reach a hundred thousand occupants before the one in New York?
**AI Predicted Answer:** Albany, Georgia is a city in the U.S. state of Georgia. Albany, Georgia has a population of 59,058. The Albany in New York has a population of 328,058. So the answer is no

**Q:** {question}
**AI Predicted Answer:** {teacher_explanation} So the answer is

**Figure 10:** Examples of StrategyQA prompts for the mental model of a teacher simulating student predictions pre-intervention and post-intervention. Pre-intervention: The demonstrations use student explanations and student predictions and at test time, the teacher simulates both. Post-intervention: The demonstrations use teacher explanations and student predictions and at test time, the teacher uses the teacher explanation to simulate the student prediction.

**Results with Cross-family Student and Teacher.** We observe that larger teacher LLMs can teach smaller student LLMs, even when they are of different model families. In Table 5, we report the results with Flan-T5 and LLaMA models as students and teachers.

| | | Intervention Budget | | | | | |
| --- | --- | --- | --- | --- | --- | --- | --- |
| Student | Teacher | 0% | 20% | 40% | 60% | 80% | 100% |
| Flan-T5-Large | Human | 58.51±2.00 | 63.75±0.43 | 66.95±2.19 | 73.94±2.77 | 78.02±2.40 | 81.95±1.65 |
| Flan-T5-XL | Human | 68.12±2.62 | 72.05±2.62 | 75.98±2.31 | 80.20±1.65 | 84.13±1.00 | 87.77±0.70 |
| Flan-T5-Large | Flan-T5-XL | 58.51±2.00 | 60.52±1.63 | 59.78±1.85 | 61.48±2.02 | 62.35±2.13 | 62.96±2.47 |
| Flan-T5-XL | Flan-T5-Large | 68.12±2.62 | 67.68±2.72 | 65.64±3.39 | 64.04±3.63 | 62.88±1.15 | 61.86±0.66 |

**Table 1:** RQ1 – Comparison of accuracy obtained with random intervention by Flan-T5 models at different intervention budgets on StrategyQA. As shown in the third row, Flan-T5-Large (student) accuracy improves by 5% with 100% intervention from Flan-T5-XL (teacher).

| | | Intervention Budget | | | | | |
| --- | --- | --- | --- | --- | --- | --- | --- |
| Student | Teacher | 0% | 20% | 40% | 60% | 80% | 100% |
| LLaMA-7B | Human | 61.13±2.72 | 63.60±4.82 | 68.85±3.52 | 73.36±2.18 | 78.45±2.55 | 81.22±1.57 |
| LLaMA-65B | Human | 77.58±2.24 | 80.34±2.65 | 82.67±2.06 | 87.48±1.96 | 89.37±0.25 | 92.86±0.50 |
| LLaMA-7B | LLaMA-65B | 61.13±2.72 | 62.29±1.53 | 64.91±0.67 | 66.08±1.76 | 68.99±3.14 | 69.43±3.41 |
| LLaMA-65B | LLaMA-7B | 77.58±2.24 | 75.83±2.24 | 72.92±2.72 | 72.92±2.26 | 70.88±0.90 | 69.14±0.66 |

**Table 2:** RQ1 – Comparison of accuracy obtained with random intervention by LLaMA models at different intervention budgets on StrategyQA. As shown in the third row, LLaMA-7B (student) accuracy improves by 8% with 100% intervention from LLaMA-65B (teacher).

# E  RQ2: Additional Results

**Results with stronger Flan-T5-XL teacher and weaker Flan-T5-Large student.** Table 6 compares different Intervention Functions on StrategyQA with Flan-T5-Large as the student and Flan-T5-XL as the teacher. These results are when the teacher has access to the gold labels. In Table 7, we

| | | Intervention Budget | | | | | |
|---|---|---|---|---|---|---|---|
| Student | Teacher | 0% | 20% | 40% | 60% | 80% | 100% |
| Flan-T5-Large | Human | 84.78±0.41 | 86.86±0.76 | 88.70±0.94 | 90.77±0.45 | 93.20±0.47 | 95.42±0.17 |
| Flan-T5-XL | Human | 92.38±0.16 | 92.52±0.20 | 92.43±0.28 | 92.23±0.61 | 92.41±1.12 | 92.21±1.06 |
| Flan-T5-Large | Flan-T5-XL | 84.78±0.41 | 85.79±0.48 | 86.79±0.84 | 87.46±0.20 | 88.52±0.39 | 89.72±0.68 |
| Flan-T5-XL | Flan-T5-Large | 92.38±0.16 | 90.92±0.39 | 89.74±0.39 | 87.98±0.89 | 86.70±1.60 | 85.19±1.62 |

**Table 3:** RQ1 – Comparison of accuracy obtained with random intervention by Flan-T5 models at different intervention budgets on CommonsenseQA. As shown in the third row, Flan-T5-Large (student) accuracy improves by 5% with 100% intervention from Flan-T5-XL (teacher).

| | | Intervention Budget | | | | | |
|---|---|---|---|---|---|---|---|
| Student | Teacher | 0% | 20% | 40% | 60% | 80% | 100% |
| LLaMA-7B | Human | 9.62±1.53 | 11.97±0.80 | 13.84±1.02 | 16.32±0.57 | 18.72±0.78 | 21.05±0.65 |
| LLaMA-13B | Human | 16.45±1.80 | 18.44±2.16 | 20.34±1.60 | 22.41±2.46 | 24.91±2.07 | 26.88±2.34 |
| LLaMA-7B | LLaMA-13B | 9.62±1.53 | 10.20±1.06 | 10.68±0.82 | 11.24±0.50 | 11.92±1.15 | 12.25±0.94 |
| LLaMA-13B | LLaMA-7B | 16.45±1.80 | 15.87±1.62 | 15.56±1.44 | 14.88±1.89 | 14.68±1.88 | 14.27±1.70 |

**Table 4:** RQ1 – Comparison of accuracy obtained with random intervention by LLaMA models at different intervention budgets on GSM8k. As shown in the third row, LLaMA-7B (student) accuracy improves by 3% with 100% intervention from LLaMA-13B (teacher).

| | | Intervention Budget | | | | | |
|---|---|---|---|---|---|---|---|
| Student | Teacher | 0% | 20% | 40% | 60% | 80% | 100% |
| Flan-T5-Large | LLaMA-65B | 58.51±2.00 | 61.86±0.25 | 61.13±2.26 | 64.48±1.53 | 66.52±4.05 | 66.95±4.90 |
| LLaMA-65B | Flan-T5-Large | 77.58±2.24 | 74.52±1.76 | 71.47±0.90 | 67.68±2.00 | 64.62±2.00 | 62.15±1.76 |

**Table 5:** RQ1 – Comparison of accuracy obtained with random intervention on StrategyQA when the student and the teacher belong to different model families. As shown in the first row, Flan-T5-Large (student) accuracy improves by 8% with 100% intervention from LLaMA-65B (teacher).

| | Intervention Budget | | | | | |
|---|---|---|---|---|---|---|
| Intervention Function | 0% | 20% | 40% | 60% | 80% | 100% |
| Random | 58.51±2.00 | 60.40±1.76 | 61.13±2.65 | 60.98±1.09 | 64.33±4.54 | 62.96±2.47 |
| Teacher Conf ↑ | 58.51±2.00 | 58.66±2.40 | 60.11±2.90 | 57.35±3.30 | 61.42±3.91 | 62.96±2.47 |
| Expected Student Conf (Pre) ↓ | 58.51±2.00 | 64.19±2.00 | 66.66±0.25 | 66.81±1.57 | 65.35±2.40 | 62.96±2.47 |
| Expected Student Conf (Post) ↑ | 58.51±2.00 | 64.77±1.76 | 68.26±0.66 | 69.71±2.01 | 68.26±2.63 | 62.96±2.47 |
| Expected Utility ↑ | 58.51±2.00 | 67.83±1.53 | 71.32±1.33 | 71.17±1.15 | 69.86±2.43 | 62.96±2.47 |
| True Student Conf (Pre) ↓ | 58.51±2.00 | 68.26±1.65 | 80.20±1.26 | 74.38±2.84 | 68.55±3.88 | 62.96±2.47 |
| True Student Conf (Post) ↑ | 58.51±2.00 | 65.64±1.40 | 72.63±1.09 | 80.05±0.90 | 72.19±4.39 | 62.96±2.47 |
| True Utility ↑ | 58.51±2.00 | 76.56±0.50 | 80.78±1.15 | 81.51±1.76 | 78.60±3.29 | 62.96±2.47 |

**Table 6:** RQ2 – Comparison of different Intervention Functions with a Flan-T5-Large student and a Flan-T5-XL teacher on StrategyQA. The teacher assumes access to gold labels. ↑ denotes that the samples are ranked in decreasing order of the function (higher is better), while ↓ denotes that the samples in increasing order of the function (lower is better).

| | Intervention Budget | | | | | |
|---|---|---|---|---|---|---|
| Intervention Function | 0% | 20% | 40% | 60% | 80% | 100% |
| Random | 58.51±2.00 | 60.40±1.76 | 61.13±2.65 | 60.98±1.09 | 64.33±4.54 | 62.96±2.47 |
| Least Conf ↓ | 58.51±2.00 | 61.13±0.75 | 62.44±1.74 | 65.06±1.15 | 63.46±2.97 | 62.96±2.47 |
| Expected Student Conf (Pre) ↓ | 58.51±2.00 | 62.59±1.00 | 61.86±0.90 | 62.29±1.33 | 65.50±3.14 | 62.96±2.47 |
| Expected Student Conf (Post) ↑ | 58.51±2.00 | 61.86±1.96 | 62.88±1.74 | 61.71±3.39 | 60.11±4.62 | 62.96±2.47 |
| Expected Utility ↑ | 58.51±2.00 | 62.29±0.50 | 62.44±1.50 | 62.44±3.88 | 62.95±2.78 | 62.96±2.47 |

**Table 7:** RQ2 – Comparison of different Intervention Functions with a Flan-T5-Large student and a Flan-T5-XL teacher on StrategyQA. The teacher, in this case, does not have access to gold labels.

| | Intervention Budget | | | | | |
|---|---|---|---|---|---|---|
| Intervention Function | 0% | 20% | 40% | 60% | 80% | 100% |
| Random | 68.12±2.62 | 67.68±2.72 | 65.64±3.39 | 64.04±3.63 | 62.88±1.15 | 61.86±0.66 |
| Expected Student Conf (Pre) ↓ | 68.12±2.62 | 66.22±2.24 | 66.95±1.53 | 65.35±1.00 | 62.73±0.66 | 61.86±0.66 |
| Expected Student Conf (Post) ↑ | 68.12±2.62 | 70.59±3.27 | 71.76±3.63 | 72.48±2.86 | 69.86±2.62 | 61.86±0.66 |
| Expected Utility ↑ | 68.12±2.62 | 70.88±3.27 | 71.90±2.84 | 72.63±2.24 | 68.99±1.15 | 61.86±0.66 |
| True Student Conf (Pre) ↓ | 68.12±2.62 | 74.23±3.73 | 76.27±1.40 | 68.55±1.00 | 64.04±0.90 | 61.86±0.66 |
| True Student Conf (Post) ↓ | 68.12±2.62 | 70.16±3.27 | 73.94±1.76 | 80.05±1.65 | 71.32±1.09 | 61.86±0.66 |
| True Utility ↑ | 68.12±2.62 | 79.91±2.00 | 80.93±2.06 | 80.64±2.24 | 78.16±2.00 | 61.86±0.66 |

**Table 8:** RQ2 – Comparison of different Intervention Functions with a smaller teacher (Flan-T5-Large) and a larger student (Flan-T5-XL) on StrategyQA. The teacher assumes access to gold labels.

| | Intervention Budget | | | | | |
|---|---|---|---|---|---|---|
| Intervention Function | 0% | 20% | 40% | 60% | 80% | 100% |
| Random | 61.13±2.72 | 62.29±1.53 | 64.91±0.67 | 66.08±1.76 | 68.99±3.14 | 69.43±3.41 |
| EU (w/ teacher answers) | 61.13±2.72 | 66.22±2.63 | 67.39±2.40 | 69.28±1.76 | 70.59±3.81 | 69.43±3.41 |
| EU (w/ gold label) | 61.13±2.72 | 66.52±3.27 | 70.16±0.90 | 71.47±1.09 | 72.78±2.48 | 69.43±3.41 |

**Table 9:** RQ2 – Comparison of Expected Utility (with and without access to gold labels) with random intervention, involving a LLaMA-7B student and a LLaMA-65B teacher. EU = Expected Utility. Importantly, even when the teacher does not have access to the gold labels, expected utility with teacher answers (second row) leads to a statistically significant 5% improvement in student accuracy ($p = 0.02$) at 20% intervention.

| | Intervention Budget | | | | | |
|---|---|---|---|---|---|---|
| Intervention Function | 0% | 20% | 40% | 60% | 80% | 100% |
| Random | 84.79±0.41 | 85.79±0.48 | 86.79±0.84 | 87.46±0.20 | 88.52±0.39 | 89.72±0.68 |
| Expected Student Conf (Pre) ↓ | 84.79±0.41 | 84.57±0.69 | 86.35±0.73 | 87.99±0.87 | 89.51±0.82 | 89.72±0.68 |
| Expected Student Conf (Post) ↑ | 84.79±0.41 | 86.66±0.37 | 88.69±0.19 | 90.76±0.06 | 92.43±0.61 | 89.72±0.68 |
| Expected Utility ↑ | 84.79±0.41 | 87.34±1.09 | 89.33±0.55 | 90.27±0.40 | 91.30±0.22 | 89.72±0.68 |
| True Student Conf (Pre) ↓ | 84.79±0.41 | 92.03±0.19 | 91.70±0.04 | 91.03±0.34 | 90.27±0.41 | 89.72±0.68 |
| True Student Conf (Post) ↓ | 84.79±0.41 | 87.40±0.39 | 89.59±0.53 | 92.31±0.09 | 94.98±1.57 | 89.72±0.68 |
| True Utility ↑ | 84.79±0.41 | 92.87±0.18 | 93.99±0.02 | 94.65±0.13 | 95.57±0.24 | 89.72±0.68 |

**Table 10:** RQ2 – Comparison of different Intervention Functions with a Flan-T5-Large student and a Flan-T5-XL teacher on CommonsenseQA. The teacher assumes access to gold labels.

| | Intervention Budget | | | | | |
|---|---|---|---|---|---|---|
| Intervention Function | 0% | 20% | 40% | 60% | 80% | 100% |
| Random | 9.62±1.53 | 10.20±1.06 | 10.68±0.82 | 11.24±0.50 | 11.92±1.15 | 12.25±0.94 |
| Expected Student Conf (Pre) ↓ | 9.62±1.53 | 11.11±1.44 | 11.37±1.17 | 11.56±1.34 | 12.40±1.01 | 12.25±0.94 |
| Expected Student Conf (Post) ↑ | 9.62±1.53 | 12.80±1.28 | 12.91±0.58 | 13.10±0.10 | 12.72±2.14 | 12.25±0.94 |
| Expected Utility ↑ | 9.62±1.53 | 13.68±1.87 | 14.06±1.44 | 13.99±0.80 | 13.68±0.58 | 12.25±0.94 |

**Table 11:** RQ2 – Comparison of different Intervention Functions with a LLaMA-7B student and a LLaMA-13B teacher on GSM8k. The teacher assumes access to gold labels.

compare the accuracy on StrategyQA when the teacher (Flan-T5-XL) does not have access to gold labels.

**Results with weaker Flan-T5-Large teacher and stronger Flan-T5-XL student.** RQ1 demonstrated that random intervention by a *smaller* teacher may not benefit a *larger* student. But, does Expected Utility benefit in such scenarios? We show this through Figure 8, which compares the accuracy on StrategyQA with Flan-T5-Large as the teacher and Flan-T5-XL as the student. While random intervention shows a monotonically decreasing trend with more intervention, Expected Utility improves the accuracy by 2% (68% → 70%) by paying 20% intervention cost and by 4% by paying 60% cost. Thus, we conclude that *weaker teachers can also teach stronger students with*

| Teacher Explanation Type | Intervention Budget | | | | | |
|---|---|---|---|---|---|---|
| | 0% | 20% | 40% | 60% | 80% | 100% |
| Unpersonalized-Rationales | 58.51±2.00 | 66.52±2.97 | 69.14±1.76 | 70.16±1.09 | 67.97±0.50 | 60.40±0.50 |
| Unpersonalized-CoT | 58.51±2.00 | 67.83±1.53 | 71.32±1.33 | 71.17±1.15 | 69.86±2.43 | 62.96±2.47 |
| Personalized | 58.51±2.00 | 69.28±1.26 | 71.61±1.15 | 72.63±1.33 | 68.55±1.90 | 62.73±2.80 |
| Human Explanations | 58.51±2.00 | 72.34±0.90 | 77.72±0.75 | 81.51±1.09 | 82.09±0.87 | 81.36±0.66 |

**Table 12:** RQ3 – Comparison of different kinds of teacher explanations (unpersonalized-rationales, unpersonalized-CoT, personalized, and human) on the student accuracy for StrategyQA. Here Flan-T5-Large is the student model and Flan-T5-XL is the teacher model.

| Teacher Explanation Type | Intervention Budget | | | | | |
|---|---|---|---|---|---|---|
| | 0% | 20% | 40% | 60% | 80% | 100% |
| Unpersonalized-CoT | 61.13±2.72 | 66.52±1.27 | 70.16±0.90 | 71.47±1.09 | 72.78±2.48 | 69.43±3.41 |
| Personalized | 61.13±2.72 | 68.95±1.26 | 71.86±2.72 | 72.61±1.96 | 73.17±4.00 | 69.57±1.53 |

**Table 13:** RQ3 – Comparison of unpersonalized and personalized teacher explanations on the student accuracy for StrategyQA. Here LLaMA-7B is the student model and LLaMA-65B is the teacher model.

| Demonstrations Type | #Rounds | | | | |
|---|---|---|---|---|---|
| | 1 | 2 | 3 | 4 | 5 |
| No Explanations | 55.45±2.26 | 56.04±4.19 | 58.95±4.16 | 57.35±3.21 | 57.93±2.66 |
| Student Explanations | 56.08±4.16 | 55.31±3.14 | 54.24±2.00 | 53.90±4.21 | 53.85±3.73 |
| Teacher Explanations | 55.74±2.40 | 60.84±3.71 | 59.97±2.66 | 59.82±4.55 | 61.57±1.31 |

**Table 14:** RQ4 – Results of Multi-turn interaction between the student and the teacher comparing student accuracy on unexplained test points with unexplained, student-explained and teacher-explained demonstrations.

*appropriately designed Intervention Functions, especially when the student and the teacher have some complementary benefits.*

**Results with LLaMA models.** Table 9 compares Expected Utility-based intervention with random intervention for LLaMA models (LLaMA-7B as the student and LLaMA-65B as the teacher) on StrategyQA. We evaluate expected utility in two scenarios – with and without gold labels. Both provide improvements over random intervention, as also observed with the Flan models. In particular, when the teacher does not have access to the gold labels (second row), one can compute expected utility with respect to the teacher predictions and obtain a significant 5% improvement ($p = 0.02$) in student accuracy at 20% intervention.

**Results on Other Datasets.** Table 10 compares different Intervention Functions on the Common-senseQA dataset with Flan-T5-Large as the student and Flan-T5-XL as the teacher. Table 11 reports results on the GSM8k dataset with LLaMA-7B as the student and LLaMA-13B as the teacher.

# F   RQ3: Additional Results

Table 12 compares different kinds teacher explanations on student accuracy for StrategyQA with Flan-T5-Large as the student model and Flan-T5-XL as the teacher model. Table 13 compares unpersonalized and personalized explanations on StrategyQA with LLaMA-7B as the student model and LLaMA-65B as the teacher model. Figure 12 shows five qualitative examples from StrategyQA of unpersonalized and personalized explanations generated by a LLaMA-65B teacher model for a LLaMA-7B student model. We observe a common pattern that the personalized explanations are shorter, simpler, and more directed toward answering the question. The unpersonalized explanations, while still factually correct, are elaborate (e.g., see 'Example 5') that may end up distracting the

| | Intervention Budget | | | | | |
|---|---|---|---|---|---|---|
| Intervention Function | 0% | 20% | 40% | 60% | 80% | 100% |
| Random | 58.51±2.00 | 60.40±1.76 | 61.13±2.65 | 60.98±1.09 | 64.33±4.54 | 62.96±2.47 |
| Neg Expected Utility | 58.51±2.00 | 52.98±1.76 | 51.09±1.57 | 50.80±1.40 | 53.42±3.10 | 62.45±1.57 |

**Table 15:** RQ5 – Comparison of random intervention function with negative expected utility, demonstrating that the teacher can hurt the student by intervening on samples where the utility is the lowest.

| | Intervention Budget | | | | | |
|---|---|---|---|---|---|---|
| Teacher Explanation Type | 0% | 20% | 40% | 60% | 80% | 100% |
| Unpersonalized-CoT | 58.51±2.00 | 67.83±1.53 | 71.32±1.33 | 71.17±1.15 | 69.86±2.43 | 62.96±2.47 |
| Deceiving Explanations | 58.51±2.00 | 66.95±1.76 | 69.86±1.15 | 68.70±0.66 | 66.66±1.40 | 62.73±3.27 |

**Table 16:** RQ5 – Comparison of a deceiving teacher with an unpersonalized teacher on StrategyQA with Flan-T5-Large as the student model and Flan-T5-XL as the teacher model.

student. Hence, the personalized explanations are probably easier to reason over for a comparatively weaker student, LLaMA-7B, leading to better performance.

## G  RQ4: Additional Results

Table 14 shows RQ4 results on StrategyQA with LLaMA-7B as the student and LLaMA-65B as the teacher.

## H  RQ5: Additional Details and Results

If teacher LLMs can successfully build mental models of student LLMs, a natural follow-up question regards whether communicating misleading explanations can also weaken student models. We verify that this holds, as detailed below.

**Study Design.**   This RQ explores the negative implications of both RQ2 (i.e., when to intervene) and RQ3 (i.e., how to generate teacher explanations), now with the goal of deceiving the student. First, extending our Expected Utility-based Intervention Function (RQ2), we rank samples in *increasing* order of expected utility, such that the teacher intervenes when the utility is the lowest. Here, the teacher's goal is to communicate explanations for data points where the student gives an incorrect answer by following the teacher's explanation but would have answered correctly had it leveraged its own explanation. We compare this with random teacher intervention (which is generally helpful). Next, in order to explore the negative implication of RQ3, we make the teacher condition on *incorrect* answers and *non-factual* human explanations that we manually generate by perturbing (correct) human explanations. We manually make minimal edits to the explanations such that the CoT reasoning is plausible yet non-factual (see some examples in Fig. 11). Now, the teacher's goal is to learn from non-factual explanations and generate similar explanations that purposefully mislead the student. We compare this misaligned teacher with an unpersonalized teacher that learns from factual gold explanations (i.e., the baseline from RQ3).

**Main Results.**   We show the results on StrategyQA in Table 15 and Table 16 with Flan-T5-Large as the student and Flan-T5-XL as the teacher. Ranking data points by negative expected utility allows the teacher to reduce student accuracy to 53% at 20% intervention and as low as random chance at 60% intervention (which is 8 points worse than the student performance with no intervention). Next, Fig. 8(b) illustrates that the teacher can condition on non-factual explanations to successfully generate worse explanations that reduce the student accuracy by at least 2 points, relative to the improvement of benign explanations. Interestingly, many of the explanations are still helpful on an absolute scale, though more than half of the deceiving explanations flip the student prediction to be incorrect (based

on the trend declining earlier than the standard CoT explanations). Therefore, our conclusion is that *teacher LLM explanations can be influential for student LLMs in both good and bad ways.*

| **Example 1** |
|---|
| **Q:** Would someone with back pain enjoy picking strawberries?
**Correct Explanation:** Back pain may be worsened by repeated bending at the waist. Strawberries grow very close to the ground.
**Correct Answer:** no
**Deceiving Explanation:** Back pain may be worsened by repeated bending at the waist. Strawberries grow high up on the tree. |
| **Example 2** |
| **Q:** Could a snowy owl survive in the Sonoran?
**Correct Explanation:** The Snowy owl is a bird native to the Arctic regions of North America. Temperatures in the North American Arctic range from -36.4F to 50F. The Sonoran is one of the hottest deserts in the world. The Sonoran Desert can get as hot as 118F.
**Correct Answer:** no
**Deceiving Explanation:** The Snowy owl is a bird native to the Arctic regions of North America. Temperatures in the North American Arctic range from -36.4F to 50F. The Sonoran is one of the most snowy mountains in the world. |
| **Example 3** |
| **Q:** Has Drew Carey outshined Doug Davidson's tenure?
**Correct Explanation:** Drew Carey has been the host of the Price is Right for over 13 years. Doug Davidson hosted the Price is Right from 1994-1995.
**Correct Answer:** yes
**Deceiving Explanation:** Drew Carey has been the host of the Price is Right for over 13 years. Doug Davidson is hosting the Price is Right from 1994 until now. |

**Figure 11:** Examples of manually constructed deceiving explanations for StrategyQA that support the wrong answer. The deceiving part of the explanations is highlighted in red.

| **Example 1** |
|---|

**Question:** Can a computer be programmed entirely in Boolean algebra?
**Gold Answer:** yes
**Unpersonalized Teacher Explanation:** Boolean algebra is a branch of mathematics that deals with the values true and false. A computer program is a set of instructions that a computer can execute.
**Student Prediction (with Unpersonalized Explanation):** no
**Personalized Teacher Explanation:** Boolean algebra is a branch of mathematics. It is used in computer programming.
**Student Prediction (with Personalized Explanation):** yes

| **Example 2** |
|---|

**Question:** Would Amy Winehouse's death have been prevented with Narcan?
**Gold Answer:** no
**Unpersonalized Teacher Explanation:** Amy Winehouse was a British singer who died of alcohol poisoning in 2011. Narcan is a brand name for naloxone, a drug used to treat opioid overdoses.
**Student Prediction (with Unpersonalized Explanation):** yes
**Personalized Teacher Explanation:** Narcan is a drug used to reverse opioid overdoses. Amy Winehouse died of alcohol poisoning.
**Student Prediction (with Personalized Explanation):** no

| **Example 3** |
|---|

**Question:** Could Brooke Shields succeed at University of Pennsylvania?
**Gold Answer:** yes
**Unpersonalized Teacher Explanation:** Brooke Shields is an American actress. The University of Pennsylvania is a private Ivy League university in Philadelphia.
**Student Prediction (with Unpersonalized Explanation):** no
**Personalized Teacher Explanation:** Brooke Shields attended Princeton University.
**Student Prediction (with Personalized Explanation):** yes

| **Example 4** |
|---|

**Question:** Can Planned Parenthood tell your University that you have Herpes simplex virus?
**Gold Answer:** no
**Unpersonalized Teacher Explanation:** Planned Parenthood is a non-profit organization that provides reproductive health care. Herpes simplex virus is a virus that causes herpes. Herpes is a sexually transmitted disease.
**Student Prediction (with Unpersonalized Explanation):** yes
**Personalized Teacher Explanation:** Planned Parenthood is a non-profit organization that provides reproductive health care. They do not have access to your medical records.
**Student Prediction (with Personalized Explanation):** no

| **Example 5** |
|---|

**Question:** Did US President during Spanish-American War suffer similar demise to Abraham Lincoln?
**Gold Answer:** yes
**Unpersonalized Teacher Explanation:** The Spanish-American War was fought in 1898. William McKinley was the US President during the Spanish-American War. McKinley was assassinated in 1901. Abraham Lincoln was the US President during the American Civil War. Lincoln was assassinated in 1865.
**Student Prediction (with Unpersonalized Explanation):** no
**Personalized Teacher Explanation:** William McKinley was assassinated in 1901. He was the 25th President of the United States.
**Student Prediction (with Personalized Explanation):** yes

**Figure 12:** Qualitative comparison between unpersonalized and personalized explanations generated by a LLaMA-65B teacher model for a LLaMA-7B student model for StrategyQA questions. For all these questions, the personalized explanation leads to the correct student answer but the unpersonalized one does not. A common pattern is that the personalized explanations are shorter, simpler, and more directed toward answering the question.

