# OpenReview forum: "Can Language Models Teach? Teacher Explanations Improve Student Performance via Personalization"
_NeurIPS.cc/2023/Conference — NeurIPS 2023 poster_

### Official Review · Reviewer_S1Bk · 2023-06-30

**Soundness:** 3 good
**Presentation:** 2 fair
**Contribution:** 3 good
**Rating:** 6
**Confidence:** 2

**Summary:**

This work centers around the research question: *can stronger LLMs teach weaker ones with their explanations? And if yes, when and how should the teacher do this*? It answers a few research questions, progressively building up from using a teacher LLM to generate explanations for a randomly sampled subset of the test data called *the intervention budget* (from 0 to 100% in steps of 20%), to also using the teacher LLM to select which points to generate explanations for, to additionally "personalise" the explanations by adding a few demonstrations of the student's behaviour to the prompt, and finally looking into whether the student can generalise from explanations on other examples to different test examples.
The results are positive, showing that in some cases a stronger teacher can teach a weaker model and increase performance by a couple percentage points. The authors propose an interesting notion of expected utility that is meant to estimate comparatively which test example it should generate explanations for to gain the most performance boost. The most compelling result from the paper is that using this estimated expected utility, which is only based on a few demonstrations, the teacher can improve the student much more than when it generates explanation for a random subset of the test examples.

**Strengths:**

The experimental setup mostly seems, as far as I can judge, sound and comprehensive (few-shot demonstrations used for models, 3 different seeds sampling from the models are used, 3 different reasoning datasets are used and two different family of LLMs).

As mentioned in the summary, I find the most compelling and interesting result and contribution from the paper the expected utility that can choose which test examples to generate an explanation for from only a few demonstrations.


**Weaknesses:**

1. I think the *main weakness* of this paper is how many details are left out of the paper. Based on the main text, I have many question and without the answer it's sometimes hard to judge the actual soundness of the setup and the conclusions derived from the experiments. See Questions section below. I want to stress that I think because of this weakness, I might have misunderstood the weaknesses I mention below, and they might turn out not to be weaknesses. However, because the presentation is such that this is all unclear to me even after looking at the full appendix, I feel like this needs to be changed to be able to judge the merits of this work.

2. Another weakness is that the performance increases seem very marginal. Often, the confidence intervals based on running with 3 seeds seem to overlap between no intervention and 100% intervention for RQ1, and from the appendix it seems that for LLama in fact generating explanations does not significantly help at all for some of the datasets (Figure 2 in the main paper, Figure 3 in appendix). For RQ2, you indeed see much larger improvements, but for those the teacher has access to a gold label, which I address below in point 3. Then when there's no access to gold labels (Figure 4b main text, table 8 appendix), the improvements from using the expected utility seem marginal / mostly overlapping with baseline confidence intervals. Additionally, I'm not sure I can find the results for LLama without access to gold labels? For RQ3, the ToM results from Figure 5 again seem to have overlapping confidence intervals with the unpersonalised CoT baseline. For RQ4, if I understand correctly, it seems to show that without access to gold labels for all explanations, using only a few, students can generalise performance across data points, although the numbers are hard to follow from the text (line 360-361 says two in-context examples improve student accuracy by 4% to 61% in round 1, is that without teaching? or after round 1?). But are there also results of baselines in RQ4, e.g. using unpersonalised CoT baseline?

3. The claim for RQ3 is that the model's Theory of Mind improves the explanations, because it conditions on human helpful explanations that flip the answer, and student's unhelpful explanations; but it seems like there is no baseline that only conditions on the human's helpful explanations; how can you then claim that it's theory of mind that's doing the work and not simply the human unpersonalised helpful explanations? Additionally, to claim that the model's theory of mind skills help improve explanations here is a bit strong, because I have seen no paper claim that LLama or Flan-T5 has theory of mind. The papers you refer to (5 and 43) only show that RLHF-ed models have ToM skills and LLama and Flan-T5 are not RLHF-ed. Further, other papers have shown that [5] doesn't use a valid experimental protocol (https://arxiv.org/abs/2302.08399) and yet others have shown that even from the RLHF-ed models, the ToM skills are not robust (work by Maarten Sap et al, e.g. https://arxiv.org/abs/2305.14763).

3. Another weakness is that the limitations section is entirely written in the appendix, and it doesn't talk about the fact that the teacher model has to have access to the gold label for explanations; this seems to me like the most significant limitation of this method. It's hard to imagine what kind of applications you can use this method if you always need the ground-truth label for examples to get the student to improve.

**Questions:**

**In summary**, I think I can see the merit of this method when the teacher does not use gold labels for each explanation, but it's hard to judge from the current presentation of the results whether that does indeed lead to a better performance than baselines across datasets and models.

Below I'll ask questions surrounding my three points of weaknesses mentioned above.

1. Many details left out, making it hard to judge the soundness
- How do you know that the explanations of the teacher model never contain the ground-truth answer? Do you check this manually, or some other way?
- You say that the student model is weaker than the teacher; how do you measure this or is it a (fair) assumption based on the model size? I would explicitly state that.
- Why do you put all results for LLama in the appendix when you do mention it like it's part of the main evaluations in the experimental setup section?
- Most important question: what does it mean for the teacher to predict the confidence of the student on an example pre- and post-explanation? Do you ask the language model to output a float / number between 0-100? Or do you make predictions with the teacher as if it were the student? Right now, it's unclear from the main text.
- In general; what do you use as a confidence for the baselines in RQ2? Is it based on the likelihood the model assigns to the answer?
- Which model generates the CoT explanations for the baseline in RQ3?

2 & 3. performance increase marginal, and some claims seem not properly backed by ablations / baselines.
Are the differences in performance between your method and the baselines significant? Can you summarise in the main text how much of an improvement your method usually gives, e.g. average improvement on the 3 datasets and for both model families? Can you show that the performance increase in RQ3 is really due to a combination of human helpful explanation and model unhelpful explanation and not simply because of the human helpful explanation?

4. Can you talk more in the main text about what this method could be helpful for given that it relies on availability to both a larger model and the ground-truth answer?
Or alternatively, can the experiments showing that it also works without ground-truth answer get more space in the main text, and also be done for all datasets and model families?

**Limitations:**

I would like to see more discussion in the main text of the limitation surrounding the need for (1) a larger model available for explanations at inference-time, and (2) the need to always have access to the ground-truth for each test example that gets explained. Re: (2), you do address this in Figure 4(b), but then seemingly do use ground-truth answers for the other RQ's. You also address (2) with RQ4, but again here the improvements seem very marginal and the section on this is very short, without baselines/ablations of the parts of the method used here (like utility). In summary, I feel like more of the paper should've been focused on these realistic setups where there is no access to ground-truth for each test example.

---

> ### Author Rebuttal · Authors · 2023-08-09
>
> We thank the reviewer for their thorough review, acknowledging our experimental setup and finding the result on Expected Utility compelling. Below we address your questions and comments.
>
> > **Many details are left out of the paper**
>
> Please refer to the common response and detailed answers here. We had to leave some details in the appendix due to space constraints. We’ll also add some of these clarification points to the main paper. If our new results and responses are convincing, we hope that you can adjust your review and scores accordingly.
>
> > **Performance increases seem very marginal**
>
> As addressed in the common response, some of our marginal gains are only down to our choice of the teacher model, which wasn’t that much stronger than the student (LLaMA-7B w/ LLaMA-13B and Flan-T5-Large w/ Flan-T5-xl). But now we address this by conducting experiments with a stronger teacher in LLaMA-65B (which for an academic lab, is quite time and resource intensive).
>
> > **Results for LLama without access to gold labels?**
>
> Please refer to our common response again for this. In summary, even without access to gold labels, expected utility improves student accuracy by up to 4 points at 20% budget. This happens automatically because the teacher is now stronger and hence a more accurate teacher.
>
> > **The claim for RQ3 is that the model's Theory of Mind improves the explanations but it seems like there is no baseline that only conditions on the human's helpful explanations**
>
> Please note that the ToM inspired teacher prompt only contains helpful human explanations that flip the student answer (refer to the ToM prompt in Fig. 1). There are no student’s unhelpful explanations. Hence, the most pertinent baseline to compare this with is using a teacher that conditions on random human explanations (which we refer to as an unpersonalized teacher). We find that a teacher that conditions on helpful human explanations leads to some improvement in student accuracy. That said, it’s true that the ToM in the models is weak, but we do have some examples of positive evidence.
>
> > **I have seen no paper claim that LLama or Flan-T5 has ToM**
>
> We are also not aware of past work arguing for ToM in LLama or FLAN-T5 in particular, but our experiments provide some evidence precisely for this conclusion. Given the increases in student performance we observe, we have evidence that the teacher builds a mental model of the student by virtue of the data provided in the prompt. If the teacher did not adjust its explanations to the student’s needs, we would not have seen any improvements over unpersonalized explanations (see General Response RQ4). So we think we should take an optimistic stance on the teacher having some weak form of ToM, although we should admit that the ToM may be extremely limited and not like human ToM. This conclusion would be consistent with prior work arguing that some LLMs are beginning to display some ToM ability (see citation [5] in our paper).
>
> > **You always need the ground-truth label for examples to get the student to improve.**
>
> As shown in experiments with LLaMA-65B (Research Question 2 -> Table 2, 2nd row), we do not need gold answers for the student to improve. If we consider the teacher models to be even stronger, like a GPT-4, then it’s not hard to imagine it obtaining 90%+ accuracy on most tasks. This effectively means that one can compute expected utility with the teacher’s predictions and not observe any significant performance drop.
>
> > **How do you know that the explanations of the teacher model never contain the ground-truth answer?**
>
> Yes, this never happens (and we manually verify 100 samples too) because the human explanations are background facts and having them as in-context examples also prompt the model to generate similar fact-based explanations in a CoT (“[Explanation]. So the answer is [answer]”). So we can just extract the explanation from this.
>
> > **You say that the student model is weaker than the teacher; how do you measure this?**
>
> In all experiments, the model with more parameters (teacher) obtains a higher accuracy than the model with lesser parameters (student). E.g., in StrategyQA, without any CoT prompting, LLaMA-7B obtains 56% accuracy while LLaMA-65B obtains 67% accuracy. Hence, we think that it’s fair to consider the bigger model with more parameters as the stronger model.
>
> > **Why all results for LLama in the appendix?**
>
> This is only due to space constraints. For better readability, we just chose the Flan-T5 models and studied all research questions with it for the main paper. The common response now shows all LLaMA results.
>
> > **What does it mean for the teacher to predict the confidence of the student on an example pre- and post-explanation?**
>
> As you understood correctly, the teacher makes predictions, as if it were the student. The prompts in Fig. 1 should help clarify this. The teacher basically learns to simulate the student by conditioning on the appropriate demonstrations and then generates an answer to the question (e.g., yes/no for StrategyQA). Then we take the likelihood of the correct answer as the confidence.
>
> > **What do you use as a confidence for the baselines in RQ2?**
>
> Yes, as per your understanding, we compute confidence as the likelihood the model assigns to the correct answer. We take the logits from the last layer of the model, and normalize it to get the correct answer's probability.
>
> > **Which model generates the CoT explanations for the baseline in RQ3?**
>
> The teacher model, which here is a Flan-T5-xl model. Note that the backbone model remains unchanged between personalized and unpersonalized teachers, only the prompt changes.
>
> > **What this method could be helpful for?**
>
> Please refer to the common response where we describe concrete use-cases of our research.

---

> > ### Comment · Reviewer_S1Bk · 2023-08-11
> > **Thanks for the rebuttal**
> >
> > Thank you for the detailed rebuttal with the new results using LLaMa-65B. Below are my responses. The summary is that the results shown with LLaMa-65B are indeed more convincing, but I still think you should refrain from referring to "theory of mind" at all in your work. Additionally, I'd like for the authors to be more explicit on what details they will add to the main paper, because as it stands it was too difficult to judge the experimental setup from the paper.
> >
> > > _"If we consider the teacher models to be even stronger, like a GPT-4, then it’s not hard to imagine it obtaining 90%+ accuracy on most tasks"_
> >
> > I don't understand why the authors make this claim in the rebuttal as the truth of this claim is completely unclear since GPT-4 is not used at all in this paper.
> >
> > > _"This conclusion would be consistent with prior work arguing that some LLMs are beginning to display some ToM ability (see citation [5] in our paper)."_
> >
> > As mentioned in my original review, [5] shouldn't be seen as evidence for anything because the experimental setup is flawed. See [this work](https://arxiv.org/abs/2302.08399).
> >
> > > _"Yes, this never happens (and we manually verify 100 samples too)"_
> >
> > I still think you should caveat this in the paper, after checking 100 samples + extracting the explanation you still cannot be fully sure the answer is not also in the explanation. For example, in my own CoT work I've very often found that the model is repeating the answer at the end which is also part of the explanation.
> >
> > > _"So we think we should take an optimistic stance on the teacher having some weak form of ToM"_
> >
> > I respectfully disagree. What you've shown is that a teacher can take examples of a student's knowledge and adjust a prompt such that that student gets a better performance after. This is personalisation. Personalisation is not the same as Theory of Mind, as reviewer UsZZ also mentions. Naming what you find in this paper theory of mind really does a disservice to the field that is actually carefully looking at theory of mind in humans and machines. That field very carefully constructs control tasks to be sure that the behaviour tested can actually fall under theory of mind, which is the ability to reason about unobserved mental states of others. Your teacher model might be reasoning only about observed mental states of others, from the prompt. Additionally, the field looking into ToM usually investigates whether the behaviour also shows up when the object being reasoned over is not an agent, but inanimate. If that is the case, it can also not be called theory of mind. This is not done at all in this work. I urge the authors to look at works in cognitive science and AI to see examples of protocols that sufficiently rule out the possibility of some behaviour not being theory of mind, but another type of reasoning (e.g. Woodward (1998), Ullman (2023), Shapira et al. (2023)).
> >
> > **What I would like to see in the main paper to judge the soundness of the protocol**:
> >
> > - How the teacher estimates the confidence of the student
> > - How you are sure that the explanations never contain the ground-truth
> > - What exactly the measurement of a stronger LLM is over a weaker one
> > - Which model generates the CoT explanations in RQ3
> > - How you compute the confidence for the baselines in RQ2
> > - Being more explicit about the results that do not need a ground-truth label
> >
> > I'm happy to raise my score if the authors can be more explicit about how they will make sure the soundness of the protocol can be judged from the main text, and if they can refrain from saying the teacher has ToM (and adjust language in the paper accordingly as well as the title) or address my points above and convincingly show that it is in fact ToM.

---

> > > ### Author Response · Authors · 2023-08-14
> > > **Replies to suggested changes in wording and clarity**
> > >
> > > Thank you for your detailed reply and for acknowledging our new results. We are happy to hear that you find these new results more convincing and are willing to increase your score. Your additional comments are also well-received.
> > >
> > > > **I'm happy to raise my score if the authors can be more explicit about how they will make sure the soundness of the protocol can be judged from the main text.**
> > >
> > > Based on the changes you suggested, please find the following clarifications with exact wordings and pointers to appropriate places where we’ll add them in the final version of the paper. Hope these satisfy all your suggestions.
> > >
> > > * How the teacher estimates the confidence of the student
> > >
> > > Clarification (after Line 249): The teacher computes these confidence estimates, as if it were the student (refer to Fig. 1 for the prompts). The teacher essentially learns to simulate the student by conditioning on the appropriate demonstrations and then generates an answer to the question (e.g., yes/no for StrategyQA). Then the likelihood of the correct answer is chosen as the simulated student confidence by the teacher.
> > >
> > > * How you are sure that the explanations never contain the ground-truth
> > >
> > > Clarification (after Line 172): We also manually verify (up to 100 samples) that the explanations do not explicitly leak the answer. This is also indirectly verified later in the experiments with the observation that the student accuracy, even with 100% teacher intervention, is far from perfect (which, if happens, may suggest answer leakage).
> > >
> > > * Which model generates the CoT explanations in RQ3
> > >
> > > Clarification (after Line 330): Note that across all the methods in this study, the explanations are generated by the teacher model (Flan-T5-XL in this case), allowing us to study the effect of the structure of the teacher prompt (rationales versus CoT) and the choice of examples in it (unpersonalized versus personalized) on the explanation generation quality.
> > >
> > > * What exactly the measurement of a stronger LLM is over a weaker one
> > >
> > > Clarification (after Line 119): Following past work, an LLM with more parameters is considered a stronger model due to its better performance across a range of tasks (including the ones we consider in our studies).
> > >
> > > * How you compute the confidence for the baselines in RQ2
> > >
> > > Clarification (after line 226): We compute model confidence as the likelihood the model assigns to the correct answer. We take the logits from the last layer of the model, and normalize it to get the correct answer's probability.
> > >
> > > * Being more explicit about the results that do not need a ground-truth label
> > >
> > > Clarification at the end of Section 4.2: To further verify this, we also conduct experiments with a stronger teacher (LLaMA-65B) which is a more accurate teacher, while using LLaMA-7B as the student. We observe a statistically significant 4 points improvement in student accuracy at 20% budget (p = 0.02), even when the teacher does not have access to gold labels. In general, when leveraging a teacher that is significantly stronger than the student and hence, more accurate, computing Expected Utility with teacher answers will work well in practice, bridging the gap to results with access to gold answers.
> > >
> > > [Add LLaMA-65B results in a separate plot]
> > >
> > > > **they can refrain from saying the teacher has ToM (and adjust language in the paper accordingly as well as the title)**
> > >
> > > We understand your concern and we are happy to rephrase parts of the paper, as per your suggestion. These are the broad list of changes we’ll do:
> > > * Remove Theory of Mind from the title.
> > > * Change “Theory of Mind prompt” everywhere with a “Student Personalization prompt”.
> > > * Focus only on personalization for RQ3 and compare unpersonalized versus personalized explanations (which we already do, when naming our methods)
> > > * We want to keep some of the motivation around Theory of Mind in the introduction but then say that RQ3’s teacher prompt design is Theory of Mind-inspired (as you and Reviewer UsZZ suggested). We will also add the references that you and Reviewer UsZZ suggested from studies in cognitive science to hopefully encourage more future work in this space.

---

> > > > ### Comment · Reviewer_S1Bk · 2023-08-14
> > > >
> > > > Thanks for this. My points are adequately addressed.

---

### Official Review · Reviewer_UsZZ · 2023-07-06

**Soundness:** 3 good
**Presentation:** 4 excellent
**Contribution:** 4 excellent
**Rating:** 7
**Confidence:** 4

**Summary:**

The authors investigate whether language models make good teachers for other, weaker language models. They do this through a systematic investigation of a teacher LM intervening at test time to improve a student LMs performance. They take seriously the idea that the teacher intervention imposes some cost (primarily on teacher, but also sometimes on the student) which then raises the question of whether teacher LMs can intervene efficiently. They dig deeper to see whether a teacher LM could personalise their explanations using a Theory of Mind based prompt, and whether teacher explanations generalize beyond the explained examples. Overall, they find positive evidence for all of these question, though with varying strengths, and introduce interesting new questions raised by this work.


**Strengths:**

Overall, I thought this paper was highly original and refreshingly systematic, building a picture not only of Teacher LM capacities but generally how we should structure our own scientific thinking about teaching competencies in LMs. It was a pleasure to read, and I think will inspire new research in this domain.

**Weaknesses:**

I think the biggest weakness, which I see in too many ML evaluation papers, is that there doesn’t appear to be any statistical analysis of the results. Generally speaking, we don’t care about the particular set of prompts used, or these particular random seeds, but want to know if the results to the -- admittedly, well thought-out -- research questions generalize beyond the the details of these evaluations (eg., the random seeds used). These are well-designed studies with nice structure and control conditions, so it would make sense to construct multi-level (aka hierarchical or mixed-effects) logistic regression models predicting the correct choice as a function of intervention budget and teacher-student relationship (which varies across RQ). See See Lampinen et al. (2022) _Can language models learn from explanations in context?_ for an example of this kind of analysis in the language model evaluation context, or see Gelman and Hill (2006) for further background.

I found the language used to describe Theory of Mind was a little strong and oversold. It seems to me the authors take “having a Theory of Mind” and “having a Theory of Mind inspired prompt” to mean the same thing, when they are not. There are many papers in computational cognitive science that take seriously important features about Theory of Mind (e.g., that it is a “theory” and has certain structural properties) that would be valuable to engage with. Please engage with these and be more nuanced in your use of language around LLM capacities, especially around what it means to have a ToM
- Baker, C. L., Saxe, R., & Tenenbaum, J. B. (2009). Action understanding as inverse planning. Cognition, 113(3), 329-349.
- Jara-Ettinger, J., Gweon, H., Schulz, L. E., & Tenenbaum, J. B. (2016). The naïve utility calculus: Computational principles underlying commonsense psychology. Trends in cognitive sciences, 20(8), 589-604.
- Baker, C. L., Jara-Ettinger, J., Saxe, R., & Tenenbaum, J. B. (2017). Rational quantitative attribution of beliefs, desires and percepts in human mentalizing. Nature Human Behaviour, 1(4), 0064.

Also, there’s a difference between Theory of Mind and personalisation: One can have a ToM, but operate under bad assumptions about the other person’s mental state, and not be able to personalise very well.

The authors talk about the trade-offs in teaching very nicely, and there is work in Cognitive Science concerning the same issues, and studying them in young children that I think the authors will find resonant.
- Bridgers, S., Jara-Ettinger, J., & Gweon, H. (2020). Young children consider the expected utility of others’ learning to decide what to teach. Nature human behaviour, 4(2), 144-152.

Some small points about Figure 1
- Might be easier to orient from Left to Right, with the Student on the left, for readability.
- A little bit confusing about what is an "answer" and what is an "explanation". In the text, it seems like the explanation is an actual informative explanation, but here it makes it seem like it's just the answer.

**Questions:**

- The authors mention a ‘human teacher’ in many of comparisons in the paper, but no mention of who or what this ‘human teacher’ is. Please clarify.
- The paper has several nicely consistent plots except for the y-axis, which makes it hard to really take advantage of the commonality of the designs and draw inferences. As well, scaling the y-axis on plots is generally bad practice as it can be misleading. Please unify the scale the y-axis.
- In the Single Round Intervention, is there a reason to not also have a setup where the student generates only a prediction without an explanation?
- The language at the end of the Single Round Intervention, talks about “When the teacher is a human, intervention happens with a human-written explanation.“, but in above it says that “e^i_H is always a human-written explanation”.
- On page 7, How does Expected Utility compare to True Utility? I may have missed something but does the teacher get the student confidence when it is not communicated? And if so, how?
- Also in the paragraph, the sentence “Ranking by teacher confidence is ineffective because it does not take the student into account.” reads like a tautology. Can you rephrase, or better yet, provide an example of how the teacher confidence can be misaligned from the student confidence?
- In the Teacher Theory of Mind Prompt, the teacher is steered towards explanations that help the student but not necessarily against explanations that do not help the student. I wonder if the authors have thought about if there’s a way of strengthening this manipulation by incorporating negative examples somehow.
- Under Baselines right under this, the word “rationalizations” appears when I think the authors mean “rationales”.
- The main results for RQ3 do not look like they’d be statistically significant with the appropriate statistical model. Please try to incorporate multi-level regression into your analysis so we can know where the evidence stands.
- I was little confused by the language in the intro to RQ4. The intuitive understanding of teacher explanations generalizing is showing a few explanations in the examples, and then having the student do their own CoT....Maybe this is what you’re saying, and the "that is" in the sentence tripped me up
- For the design of RQ4, Would this be the same as conditioning on the student explanations with (say) the correct answer?


**Limitations:**

I think the authors could talk more about the potential harms from excessive personalisation with LLMs.

---

> ### Author Rebuttal · Authors · 2023-08-09
>
> We thank the reviewer for their encouraging comments and detailed review, along with providing useful references. We will add them in the final version. Below we address your questions and concerns.
>
> > **Statistical analysis of results**
>
> Thanks for the suggestion! Please refer to our common response, where we’ve added results with a stronger teacher and carried out statistical tests for all results. These are conducted with a block bootstrap that resamples item-level accuracies across items and random seeds (which correspond to random model prompts), thus accounting for both prompt variance and test variance.
>
> > **Multi-level (aka hierarchical or mixed-effects) logistic regression models**
>
> Thanks for the thought! We are actually familiar with the methods but didn’t see a strong need for the hierarchical model in our current setup. Lampinen et al uses a hierarchical model to estimate per-item difficulty as a random effect based on per-task difficulty and to allow explanation effect estimates to vary by task. For now, we don’t have any multitask models but only report accuracies for individual tasks, and our control conditions always use the same k examples as our treatment conditions, so the effect sizes we estimate are attributable directly to the treatment. To be clear, our RQs1-4 do show a nesting structure, but it is somewhat artificial since our RQ4 experiment could be carried out with any combination of methods from RQ1-3 (e.g., a random utility ordering and unpersonalized + rationalized explanations). That said, we appreciate that when estimating the effect of our personalized teacher explanations on student performance in RQ4, the analysis could benefit from a hierarchical model allowing us to estimate the effect of teacher explanations at different rounds, while sharing information across rounds. At least for now, the differences between the teacher-explained condition and the two baselines are directly comparable within a particular round, and these currently reach up to 4 points based on our new results in the General Response pdf.
>
> > **I found the language used to describe Theory of Mind was a little strong and oversold**
>
> This point is well-received and we’ll adjust our wording. We agree that a prompt designed according to some ToM principle may not induce any ToM ability in an LM. However, we want to argue that our ToM inspired prompt does in fact lead to the teacher better anticipating the student’s behavior, suggesting that the teacher builds some mental model of the student. We do not know what aspects of the student are being modeled within the teacher (that is a kind of interpretability problem), but given the small increase in student performance, we have some positive evidence that the teacher builds a mental model of the student by virtue of the data provided in the prompt. So we are taking an optimistic stance on the teacher having some weak form of ToM, although we should admit that the ToM may be extremely limited and not like human ToM.
>
> > **No mention of who or what this ‘human teacher’ is.**
>
> All the explanation datasets used in our study are annotated with human-written explanations (typically, crowdsourced on platforms like Amazon Mechanical Turk in prior work). When these human explanations are being communicated to the student during intervention, we say that the teacher is a human.
>
> > **Unifying the scale in y-axis**
>
> Thanks, we’ll do this for the plots in the final version.
>
> > **Results without explanation**
>
> We did not include this because this is already studied in prior work like Chain-of-Thought. Yet, for the sake of completeness, we are reporting the LLaMA-7B and LLaMA-65B results on StrategyQA below. This is consistent with prior work that CoT improves performance and bigger models benefit more.
>
> |          |    LLaMA-7B    |    LLaMA-65B   |
> |----------|:--------------:|:--------------:|
> | w/o expl | 56.04±4.19     | 67.68±3.57     |
> | w/ expl  | **61.13±2.72** | **77.58±2.24** |
>
> > **“e^i_H is always a human-written explanation”.**
>
> To clarify, e^i_H are the human explanations in the in-context examples. However, when we say that “for human teachers, intervention happens with human-written explanation”, the explanation for the test instance is also a human explanation.
>
> > **How does the teacher get the student confidence?**
>
> It’s true that student confidence is not being communicated when we are computing **expected utility**. However, to establish an upper bound for this estimate, we compute True Utility which is when we assume that the student does communicate its confidence to the teacher.
>
> >  **“Ranking by teacher confidence is ineffective because it does not take the student into account.”**
>
> When we rank the samples by teacher confidence, it only tells us about the samples that the teacher is good or bad at. This may not correlate with the capabilities of the student. Hence, we observe that ranking by teacher confidence is not a good indicator of when the student needs intervention.
>
> > **Contrastive explanations in Teacher ToM prompt**
>
> We actually tried it but did not observe statistically significant improvements with it. We believe that this is still a great suggestion and worth exploring more in future work (e.g., with even stronger teachers like GPT-4).
>
> > **RQ4 wording**
>
> RQ4 tests whether the teacher can explain a few data points that then go in the student prompt and the student conditions on them to perform their own CoT at test time without teacher’s intervention. For studying RQ4 better, we have now added additional results with a stronger teacher (LLaMA-65B) in the common response. Please refer to it and let us know if there are any follow up questions.

---

> > ### Comment · Reviewer_UsZZ · 2023-08-16
> > **Thanks**
> >
> > Thanks to the authors for their responses and clarifications. My main concern was about the ToM based language, which reviewer S1Bk has thoughtfully championed. I see that you are going to remove the strong ToM language from the paper in service of a more accurate Personalization approach, which I think is a good move. Some small comments that I hope you will clean up:
> >
> > > All the explanation datasets used in our study are annotated with human-written explanations (typically, crowdsourced on platforms like Amazon Mechanical Turk in prior work). When these human explanations are being communicated to the student during intervention, we say that the teacher is a human.
> >
> > Okay, you should mention this in the main text of the paper. If you describe “human explanations” but don’t explain who the human(s) is(are), it is confusing.
> >
> > > Using a stronger teacher also helps improve the student accuracy by 2.4% (p-value = 0.09)
> >
> > Just so you know, since you are not a priori stating a p-value threshold based on a principled statistical argument (which would be outside the scope of this work anyway), you inherit the ‘standard’ p-value threshold in the literature, which is 0.05. Since 0.09 > 0.05, that means there is not evidence that “Using a stronger teacher also helps improve the student accuracy” in RQ3 (as I surmised in my original comment). I would adjust your claims accordingly.
> >
> > > We actually tried it but did not observe statistically significant improvements with it. We believe that this is still a great suggestion and worth exploring more in future work (e.g., with even stronger teachers like GPT-4).
> >
> > Since you tried it, I think you should report it, at least in a footnote. Interested readers will want to know.

---

> > > ### Author Response · Authors · 2023-08-18
> > > **Thank you!**
> > >
> > > Thank you for your comments!
> > >
> > > > **I see that you are going to remove the strong ToM language from the paper**
> > >
> > > Yes, we’ll adjust the wording to say that RQ3 is about personalization.
> > >
> > > > **Human Explanations**
> > >
> > > Yes, we'll add the exact text, as written in the previous response.
> > >
> > > > **RQ3 claim**
> > >
> > > We agree that going by the standard p-value threshold, this is not quite significant. We’ll adjust our conclusion to say that "We have some positive evidence that LLMs can personalize their explanations to better teach the student". We’ll also add some qualitative examples to the appendix.
> > >
> > > > **Contrastive Explanations in Student Prompt**
> > >
> > > Sure, we’ll add a footnote saying that this is still something worth exploring more in future work.

---

### Official Review · Reviewer_R7Nj · 2023-07-08

**Soundness:** 2 fair
**Presentation:** 3 good
**Contribution:** 2 fair
**Rating:** 4
**Confidence:** 3

**Summary:**

In this work, the authors tackle exciting research questions on how the teacher LLM intervention, which provides explanations in natural language, can enable the student LLM to make more accurate predictions. They specifically used StrategyQA, GSM8k, and CommonSenseQA for their experiments. The direction of the research is very important, and I liked the setup of formalizing the research questions with directly provided answers. However, the experimental design needs to be explained in more detail (possibly due to a missing appendix). I would also suggest considering the addition of significance testing for the obtained results to make them more convincing. Furthermore, it would be beneficial to include a discussion on how the results differ across different datasets and why.

Minor comments and questions:
- It would be helpful to explain the notation used in Fig 1 (e.g., in the caption or mention where the notation is introduced)
- '(details in Appendix A)' appendix is not available (but mentioned in the paper), so it's hard to judge some aspects of the paper since there are a lot of references to the discussion in the appendix
- 'mental model of the student' -> that needs some explanation
- The formulation of RQ1 sounds a bit ambiguous. Do the authors mean if the teacher intervention can help?
- RQ3: the notion of personalization is clearly explained (since it's an overloaded term, I would advise making it clear)
- RQ4 formulation can be clear
- 'Hence, the teacher communicates a partial explanation (specifically, only the first step of the rationale)' -- it would be great to show an example here
- lines 177-178, am I right that 1) are Teacher models, and 2) are models used for student
- Fig 2 :caption -> what is 'A + B = A student, B teacher'?
- lines 191-201: the study design is not clear from the perspectives of how a human teacher is introduced.
- Fig 2: 'Flan-T5-XL + Flan-T5-Large' -- why the performance weakens withe more explanations?
- line 221 'communication cost' -- how would you measure communication cost?
- lines 224 - 255: How exactly confidence is estimated is unclear.
- fig 4 (a) why the accuracy goes down with more interventions.

**Strengths:**

1. Exciting research direction that needs to be explored
2. Nice structure of the paper with RQ accompanied with answers.

**Weaknesses:**

1.The discussion in the paper appears fragmented, possibly due to the missing Appendix.
2. The experimental setup is not well-explained.
3. No significant testing has been applied.
4. It could be beneficial to include links to real-world applications that can benefit from this research.

**Questions:**

I mentioned them in the minor comments section above.

**Limitations:**

The authors addressed the limitations.

---

> ### Author Rebuttal · Authors · 2023-08-09
>
> We thank the reviewer for acknowledging our exciting research questions and our experimental setup with direct answers. Below we address your questions and concerns.
>
> > **Missing appendix**
>
> Please note that our paper **does have** an accompanying appendix. As per the rules, it is uploaded as a separate supplementary pdf. Please refer to it and if there is anything in particular that is not quite clear, let us know and we’ll be happy to address it. We sincerely hope that you will adjust the scores based on our responses.
>
> >  **Significance Tests**
>
> We’ve now added significance tests for all our main claims. Please refer to our general response.
>
> > **Real-world applications**
>
> Please refer to our general response.
>
> > **Discussion on how the results differ across different datasets and why.**
>
> Due to space limitations, we had to put some results on the other datasets (GSM8k and CommonsenseQA) in the appendix (see Fig. 3, Table 3, 4, 11). We also report results with a stronger teacher (LLaMA-65B) in the common response. In summary, our claims generalize across datasets with varying strengths. E.g., Fig. 3 shows that in all datasets, teacher intervention on weaker students exhibit a monotonically increasing trend.
>
> > **Notations of Fig. 1**
>
> Fig. 1 uses the same notations as in the main text (e.g., see Lines 236-255) and we do point to the figure for each research question.
>
> > **'mental model of the student'**
>
> A mental model is a concept that refers to an agent’s understanding, knowledge, beliefs, and assumptions about how a certain system works. Our paper explores whether an LLM teacher can build such a mental model of the student, such that the teacher can then use that model to simulate student behavior on unseen data points with the goal of teaching the student. This is analogous to how human teachers are also able to teach better when they can anticipate when and how a student might require teaching.
>
> > **The formulation of RQ1 sounds a bit ambiguous.**
>
> It answers whether teacher intervention on a student's reasoning can help the students to make more accurate predictions (see Lines 187-190).
>
> > **RQ3: the notion of personalization is not clearly explained**
>
> By personalization, we mean improving the teacher’s explanations based on the particular student that’s being taught (see Lines 307-313). Typically, a teacher can just generate an unpersonalized explanation (which may or may not maximally benefit all students) or the teacher can observe the students on a few data points and then mold/personalize its explanation such that the student benefits more.
>
> > **RQ4 formulation can be clear**
>
> Please refer to our new RQ4 discussion and results in the general response. If this is clear now, we’d request you to revisit the scores.
>
> > **Example of partial explanations**
>
> We do this only for the GSM8k dataset to ensure that there is no answer leakage from the communication of explanations. We first verify that all explanations in this math dataset have at least two steps of derivation, hence it’s safe for the teacher to only communicate the first step of the explanation. For example, if the entire explanation is “Natalia sold 48/2 = 24 clips in May. Natalia sold 48+24 = 72 clips altogether in April and May. So the answer is 72”, the teacher communicates only the first step “Natalia sold 48/2 = 24 clips in May.” as the partial explanation to the student. The student’s job is to then leverage it as a hint and derive the rest of the steps to arrive at a final answer.
>
> > **lines 177-178, am I right that 1) are Teacher models, and 2) are models used for student**
>
> No, these are the two model families we experiment with. So, the teacher and student models can be (1) two differently sized LLaMA models, (2) two differently sized Flan-T5 models, or (3) one Flan-T5 and one LLaMA model. See Fig 3 and Table 5 in appendix.
>
> > **Fig 2 :caption -> what is 'A + B = A student, B teacher'?**
>
> If a plot’s legend is of the form “A + B”, then A is the student and B is the teacher. So the curve annotated with “Flan-T5-Large + human” means that Flan-T5-Large is the student while human is the teacher.
>
> > **human teacher**
>
> All the explanation datasets used in our study are annotated with human-written explanations in prior work. When these human-written explanations are being communicated to the student during intervention, we say that the teacher is a human.
>
> > **Fig 2: 'Flan-T5-XL + Flan-T5-Large' -- why the performance weakens with more explanations?**
>
> 'Flan-T5-XL + Flan-T5-Large' means that Flan-T5-XL is the student and Flan-T5-Large is the teacher. So the teacher is a weaker model than the student and this result shows that with random intervention, a weaker teacher’s explanations may not be able to improve a stronger student. This is expected because on average, the stronger student is better than the weaker teacher.
>
> > **How would you measure communication cost?**
>
> Thanks, we discuss this in the appendix (limitations section). We make a simplifying assumption that the communication cost is uniform across all samples. Non-uniform costs (e.g., measured based on the number of tokens or reasoning steps in the explanations) such that longer explanations incur larger costs is an interesting direction for future work.
>
> > **How exactly confidence is estimated is unclear**
>
> We compute confidence as the likelihood that the model assigns to the correct answer. We take the logits from the last layer of the model, and normalize it to get the correct answer's probability.
>
> > **fig 4 (a) why the accuracy goes down with more interventions**
>
> Interesting question! We discuss this in Lines 275-277. Stronger model teachers, on average, are better than the weaker student models. But this does not necessarily mean that they are always better than the student. Alternatively, the student might also be too weak to understand these explanations. Hence, for a fraction of the samples, student accuracy drops with teacher explanations.

---

> > ### Comment · Reviewer_R7Nj · 2023-08-17
> > **Thanks!**
> >
> > I have read the authors' rebuttal, I really appreciate all the new provided inputs and thoughtful explanations. Most of my questions are addressed.

---

> > > ### Author Response · Authors · 2023-08-18
> > > **Thank you!**
> > >
> > > Thank you for the kind words! We’re glad to hear that we addressed your questions and we hope you can help revisit your evaluation/scores. Please also let us know if there are any follow up questions.

---

### Official Review · Reviewer_fVpe · 2023-07-23

**Soundness:** 3 good
**Presentation:** 3 good
**Contribution:** 3 good
**Rating:** 5
**Confidence:** 4

**Summary:**

The paper studies how an LLM can be used to teach another LLM how to perform reasoning tasks by providing explanations (chain-of-thoughts). The paper starts by establishing that when the teacher LLM intervenes at test-time by providing such explanations, the student LLM accuracy is indeed improved compared to having the student rely on its own explanations (RQ1). Next, the paper considers the setup in which the intervention budget is limited (i.e., the teacher can provide explanations only for a limited number of test examples). The paper proposes a method (expected utility) for selecting which examples the teacher should provide explanations for (RQ2) and a method for generating personalized explanations for a particular student (RQ3). Both methods are based on prompting the teacher model with demonstrations of observed student predictions of historical examples. While RQ1, RQ2, and RQ3 study teacher intervention at test time, the paper lastly tests whether such interventions can be used by the student to perform better on test examples that the teacher has not provided explanations for (RQ4). The experiments are conducted on three tasks: StrategyQA, GSM8k, and CommonsenseQA with four publicly available LLMs: Flan-T5-Large, Flan-T5-XL, Llama-7B, and Llama-13B.

**Strengths:**

1. The paper introduces and discusses interesting broad research questions that are likely to motivate future research efforts: (1) Evaluating LLMs from the perspective of teaching and improving weaker agents, (2) Creating personalized explanations for teaching, and (3) Distillation approaches that identify when a student lacks understanding.

2. The paper introduces an interesting method for estimating the utility of providing an explanation to the student model by simulating student behavior with different explanations. The paper demonstrates significant gains in results (Figure 3) and an interesting gap for improvement compared to the upper (True utility).

3. The paper proposes an innovative method for generating personalized explanations via conditioning on helpful (flip the incorrect prediction to a correct one) human explanations. The results are weak/mixed, though (please see #3 in Weaknesses).

**Weaknesses:**

1. In principle, providing explanations at test time for a set of test examples and achieving accuracy gains on those examples cannot be called "teaching". Teaching has to be evaluated by the ability of the taught model to generalize to new examples that it did not receive assistance with. However, the majority of this paper (RQ1, RQ2, and RQ3) defines teaching as providing explanations on test examples which seems quite misleading to me.

2. RQ4 conduct proper evaluation for the teaching setup by measuring generalization to new examples (without teacher intervention). In that setup, the teacher intervenes in multiple rounds. At each round, the teacher chooses a few examples (two for FLAN-T5 and four for LLama) to provide teacher explanations for and use such examples as demonstrations for solving test examples. However,  the paper only compares to a zero-shot prompting baseline. It is not clear if the gain in accuracy is due to providing carefully chosen and personalized teacher explanations or it is just due to moving from zero-shot to few-shot. At least, the paper needs to compare to (1) providing unpersonalized teacher explanations for randomly chosen examples, and 2) few-shot prompting with generated student explanations for randomly chosen examples. Furthermore, the results in line 357-359 indicates that after the second round of intervention, the accuracy of the student drops/saturates. The paper does not provide any explanation for that behavior.

3. [Missing qualitative analysis of personalized explanations]. How different are they from the unpersonalized ones? The paper particularly needs to provide such analysis given that the results that support RQ3 are weak and somehow mixed (Figure 5 and Table 12 in the appendix): At 20% intervention, the gain is only 1.45% (maximum achieved gain) and at 80% intervention, unpersonalized explanations even yield better accuracy by 1.3%.

**Questions:**

1. Suggestion: One of the main aspects of this study is reducing communication cost between the teacher and the student. The paper needs to provide some real-world motivating scenarios for that requirement, and distinguish it clearly from the cost incurred by invoking the teacher model.

2. Suggestion: While establishing that CoT paradigm outperforms post-hoc rationales is interesting, including post-hoc rationales as a baseline in Figure 5 (RQ3) is not justified for the purpose of that experiment. For clarity, I suggest moving that baseline to a different part of the paper, remove it, or mention why it makes sense to include it in the context of RQ3.

3. Minor typos - Table 8 & 9 in the appendix do not mention the task name. Appendices are referenced with letters A-E while the provided appendix is using section numbers.

**Limitations:**

After the discussions and new results provided by the authors (see below), I see two main limitations of this work:

1. While personalized explanations is a key aspect of the presented approach, end-to-end results show that it only leads to marginal gains. See the authors' comment below 'More baselines and analysis'

2. As discussed with the authors, the presentation of the experiments needs to be revised significantly to distinguish the evaluation of the components (RQ1, RQ2 and RQ3) from the end-to-end evaluation of the actual teaching scenario (RQ4) that the paper aims at. The experiments need to be a lot more focused on RQ4.

---

> ### Author Rebuttal · Authors · 2023-08-09
>
> We thank the reviewer for finding value in our research questions and their ability to motivate future research. We are also encouraged by the reviewer’s appreciation of our Expected Utility-based intervention function and generating personalized explanations. Below, we address your questions and concerns.
>
> > **Teaching has to be evaluated by the ability of the taught model to generalize to new examples that it did not receive assistance with.**
>
> We agree with the reviewer that one of the core aspects of "teaching" is that teacher explanations should improve student performance on unexplained data points and this is exactly what RQ4 tests. Please refer to our additional experiments for RQ4 with a stronger teacher in the common response. We have also incorporated appropriate baselines, as per your suggestion.
>
> > **However, the majority of this paper (RQ1, RQ2, and RQ3) defines teaching as providing explanations on test examples which seems quite misleading to me.**
>
> Thanks for the suggestion! We’ll adjust the terminology and wording to say something like "With RQ1-RQ3, we test teaching **components** while with RQ4, we test **teaching** ability". That said, we still want to reiterate the motivations of RQ1, RQ2 and RQ3 and why they are still important questions to study.
>
> * First, RQ1-RQ3 form basic building blocks for RQ4 and hence should be considered as important ablations of RQ4. In the context of any form of teaching, the first two questions of interest are when and how to intervene. RQ2 and RQ3 address these questions by introducing methods for both.
> * Second, the teacher's explanations only provide additional knowledge to the student, so the student still has to reason over that knowledge to produce an answer. For tasks like GSM-8k, we do not even provide the entire explanation (so treat it more as a hint). Hence, providing explanations for test examples still evaluates a non-trivial aspect of teaching which is filling in missing spots in a student's background knowledge (QA tasks), or starting the student in the right direction to solve a problem on their own (GSM8k).
>
> > **Significance of RQ3**
>
> Please refer to our common response where we perform significance analysis of our study. We’ll add some qualitative analysis in the final version due to space constraints here. Note that if a baseline achieves higher accuracy at 80% intervention, that is by design because our goal is to push the peak of the curve toward the left (i.e., at lower budgets). The samples at the tail of the curve are those where teacher explanations do not help (either because the explanations are bad or the student is not good enough to learn from those). Hence we see that student accuracy goes down beyond a certain budget.
>
> > **One of the main aspects of this study is reducing communication cost between the teacher and the student. The paper needs to provide some real-world motivating scenarios for that requirement.**
>
> Please refer to our discussion in the general response with real motivating examples for our studies.
>
> > **While establishing that CoT paradigm outperforms post-hoc rationales is interesting, including post-hoc rationales as a baseline in Figure 5 (RQ3) is not justified for the purpose of that experiment.**
>
> This is a great point! RQ3 is about how to best generate teacher explanations such that the student benefits most. Since post-hoc explanations are also a method of generating explanations, we think it is still a meaningful baseline to have. But we agree that if the comparison is only between unpersonalized and personalized explanations, then RQ3 may not be the best place to add this result. We’ll follow your suggestion by briefly mentioning it as a tuning decision and leave the result to the appendix.
>
> > **Minor typos**
>
> Table 8 and 9 are results on StrategyQA. We’ll also fix the appendix section numbers. Thank you for the suggestions!

---

> > ### Comment · Reviewer_fVpe · 2023-08-12
> > **Thanks for the rebuttal, but the results are still not fully supporting the approach.**
> >
> > 1. I agree that it is important to evaluate the components in RQ1, RQ2, and RQ3. As suggested by the authors, adjusting the terminology and making it clear that RQ4 is the part in which the teaching ability is evaluated would make sense. However, I respectfully disagree with the second argument above about additional knowledge. *"Second, the teacher's explanations only provide additional knowledge to the student .."*. Indeed, it is an important skill that student models leverage additional knowledge/hints, but it is different from teaching the model how to solve a task.
> > 2. Thanks for the additional baselines in Table 4. However, I do not see the first baseline I suggested (unpersonalized teacher explanations for randomly chosen examples) included. Given the discussion relevant to the previous point, RQ4 is the most important experiment in this paper (evaluating all components together). With the presented results in the main paper, appendix and rebuttal pdf, we do not have any evidence yet that example selection or personalized explanations help teaching when teaching is properly evaluated end-to-end as in RQ4. The paper needs to confirm with results in RQ4 the value of example selection and personalization (ideally, separately). Please, point me to such results if I am missing them in which case I will be able to raise my score.
> > 3. Thanks for the additional results and significance tests. As the personalization is one of the central aspects of the presented approach, I was hoping to see any form of qualitative analysis that shows what is really different about personalized explanations. I think that is still needed given that gains at 20% with the 65B model as teacher are significant only with p-value=0.09 (as the new results indicate).

---

> > > ### Author Response · Authors · 2023-08-15
> > > **More baselines and analysis**
> > >
> > > Thank you for the additional comments. We also appreciate your acknowledgement of the new results.
> > >
> > > > **I do not see the first baseline I suggested (unpersonalized teacher explanations for randomly chosen examples) included.**
> > >
> > > RQ2 and RQ3 are indirectly evaluated through their respective experiments but we agree that they are not evaluated in the context of RQ4. Please find these requested results below.
> > >
> > > |                                           |      1     |        2       |        3       |        4       |        5       |
> > > |-------------------------------------------|:----------:|:--------------:|:--------------:|:--------------:|:--------------:|
> > > | Teacher Expl (ExpUti + Per) | 55.74±2.40 | **60.84±3.71** | 59.97±2.66     | **59.82±4.55** | **61.57±1.31** |
> > > | Teacher Expl (ExpUti + Unper)                | 56.00±3.23 | 60.23±1.56     | **60.21±3.41** | 59.43±3.42     | 60.78±2.41     |
> > > | Teacher Expl (Random + Unper)                     | 56.62±3.30 | 59.67±1.40     | 60.00±1.31     | 57.64±1.90     | 59.53±0.90     |
> > > | Student Expl                      | 56.08±4.16 | 55.31±3.14     | 54.24±2.00     | 53.90±4.21     | 53.85±3.73     |
> > >
> > > Below we summarize our observations:
> > > * Random teacher expl work pretty well & are significantly better than student expl.
> > > * Choosing teacher expl via Expected Utility starts to show bigger gains after more rounds (p=0.07). As one includes more demonstrations, the chances of random teacher explanations hurting the students increase. Note that the effect size has gotten larger with stronger models, and the improvement is consistent across different budgets (our statistical test only applies to a single budget value and therefore understates the consistency of the improvement), so we believe this result will hold in general.
> > > * Personalized explanations leads to only marginal gains (p=0.2). Though the gains are small & not currently statistically significant, we think that the overall evidence for personalization leans positive and has gotten stronger as we have scaled up the teacher, showing future work potential with even stronger teachers. See more analysis of personalized explanations below, as you suggested.
> > >
> > > > **Qualitative analysis that shows what is really different about personalized explanations.**
> > >
> > > Some representative examples of unpersonalized versus personalized expl below where the former leads to an incorrect student prediction while the latter leads to a correct prediction. A common pattern is that the personalized expl are shorter, simpler, & more directed toward answering the question. The unpersonalized explanations, while still factually correct, are elaborate (e.g., see Q5) and may end up distracting the student. Hence, we think that the personalized explanations are easier to reason over for a comparatively weaker student, LLaMA-7B, leading to better performance.
> > >
> > > We hope our new baselines/analysis satisfy all your suggestions and you are able to adjust your scores accordingly.
> > >
> > > ```
> > > Q1: Can a computer be programmed entirely in Boolean algebra?
> > >
> > > Ans (A): yes
> > >
> > > Unpersonalized Teacher Explanation (UE): Boolean algebra is a branch of mathematics that deals with the values true and false. A computer program is a set of instructions that a computer can execute.
> > >
> > > Student Prediction (SP): no
> > >
> > > Personalized Teacher explanation (PE): Boolean algebra is a branch of mathematics. It is used in computer programming.
> > >
> > > Student Prediction (SP): yes
> > > ```
> > > ```
> > > Q2: Would Amy Winehouse's death have been prevented with Narcan?
> > >
> > > A: no
> > >
> > > UE: Amy Winehouse was a British singer who died of alcohol poisoning in 2011. Narcan is a brand name for naloxone, a drug used to treat opioid overdoses.
> > >
> > > SP: yes
> > >
> > > PE: Narcan is a drug used to reverse opioid overdoses. Amy Winehouse died of alcohol poisoning.
> > >
> > > SP: no
> > > ```
> > > ```
> > > Q3: Could Brooke Shields succeed at University of Pennsylvania?
> > >
> > > A: yes
> > >
> > > UE: Brooke Shields is an American actress. The University of Pennsylvania is a private Ivy League university in Philadelphia.
> > >
> > > SP: no
> > >
> > > PE: Brooke Shields attended Princeton University.
> > >
> > > SP: yes
> > > ```
> > >
> > > ```
> > > Q4: Can Planned Parenthood tell your University that you have Herpes simplex virus?
> > >
> > > A: no
> > >
> > > UE: Planned Parenthood is a non-profit organization that provides reproductive health care. Herpes simplex virus is a virus that causes herpes. Herpes is a sexually transmitted disease.
> > >
> > > SP: yes
> > >
> > > PE: Planned Parenthood is a non-profit organization that provides reproductive health care. They do not have access to your medical records.
> > >
> > > SP: no
> > > ```
> > >
> > > ```
> > > Q5: Did US President during Spanish-American War suffer similar demise to Abraham Lincoln?
> > >
> > > A: yes
> > >
> > > UE: The Spanish-American War was fought in 1898. William McKinley was the US President during the Spanish-American War. McKinley was assassinated in 1901. Abraham Lincoln was the US President during the American Civil War. Lincoln was assassinated in 1865.
> > >
> > > SP: no
> > >
> > > PE: William McKinley was assassinated in 1901. He was the 25th President of the United States.
> > >
> > > SP: yes

---

> > > > ### Author Response · Authors · 2023-08-18
> > > > **Gentle Reminder**
> > > >
> > > > Dear reviewer, this is a gentle reminder of our additional response above to your comments. Since we are nearing the end of the discussion period, we would very much appreciate it if you could kindly let us know if this addresses your questions (or feel free to ask us any remaining questions). Thanks again for the discussion!

---

> > > > > ### Comment · Reviewer_fVpe · 2023-08-18
> > > > > **Thanks**
> > > > >
> > > > > Thanks for the new results and qualitative examples. All my concerns are addressed at this point. I am just left with the concern that personalized explanations only leads to marginal gains. Still, the approach is very interesting, and the paper is quite inspiring. I updated my score and the limitations section above accordingly.

---

### Author Rebuttal · Authors · 2023-08-09

> **Experiments with a stronger teacher and significance of results**

Some reviewers were curious about the statistical significance of our results. Since experiments with larger teachers are significantly more time and resource intensive, we had to choose a teacher model that while stronger than the student model, wasn’t significantly stronger than the student (e.g. LLaMA-13B vs LLaMA-7B or Flan-T5-xl vs Flan-T5-Large). Hence, the margin of improvements was always going to be smaller compared to using an even stronger teacher. However, all claims made in the paper are sound, and moreover we now provide even stronger evidence by (1) scaling up – using a LLaMA-65B model as the teacher and a LLaMA-7B model as the student, keeping everything else same as in the paper and (2) adding significance tests using a non-parametric bootstrap test. All results are on the StrategyQA dataset. We are reporting the main takeaways here. Please refer to the rebuttal pdf for the tables.

**Research Question 1:** Here, the teacher randomly intervenes on student reasoning. We compare (1) LLaMA-65b teacher w/ LLaMA-7B student, and (2) vice versa. See Table 1 in rebuttal pdf. We observe about 8 points improvement in student accuracy with teacher intervention (statistically significant with p = 0.01). Hence, our RQ1 claim is further validated.

**Research Question 2:** We compare random intervention with Expected Utility. See Table 2 in rebuttal pdf. Since Reviewer S1Bk was also curious about the teacher having access to gold labels, we evaluate expected utility in two scenarios – with and without gold labels. Both improve over random intervention, further solidifying our RQ2 claims. Stronger teachers like LLaMA-65b are significantly better at solving the task, thus their predicted labels will mostly match the gold labels. Hence, even if we rely on the teacher’s predictions for computing utility (Table 2 second row), it improves by up to 4 points (statistically significant with p = 0.02) and closes the gap between the w/ and w/o gold label setups.

**Research Question 3:** We now compare unpersonalized explanations to personalized ones. See Table 3 in rebuttal pdf. Personalized explanations are generated using a ToM-inspired prompt that only conditions on helpful human explanations. Using a stronger teacher also helps improve the student accuracy by 2.4% (p-value = 0.09). Unlike our previous results, we have weaker evidence for personalization but this is still promising for further exploration with even stronger teachers.

**Research Question 4:** Finally, RQ4 tests whether teacher explanations improve students on future unexplained samples. See Table 4 in rebuttal pdf. Here, we also increase the number of teaching rounds to 5, adding 2 explained examples in each round. As recommended by Reviewer fVpe and UsZZ, we compare teacher-explained demonstrations with two baselines: (1) demonstrations with no explanations (i.e., only QA) and (2) demonstrations with self-explanations (i.e., demonstrations with student explanations + predictions). Since we study the usefulness of teacher explanations, we keep the in-context examples unaltered in this setup. Below, we summarize the conclusions.

* We observe (in Table 4, last row) that teacher explanations improve student performance on future unexplained test points as well by a significant 6 points (55% → 61.6%).
* While student accuracy with unexplained samples improves after adding more demonstrations (55% -> 59%), accuracy with teacher-explained demonstrations is better by up to 4 points (p-value = 0.04) after each round.
* When the student conditions on self-explained demonstrations (Table 4, second row), its performance decreases (56% -> 54%) with increasing rounds. We hypothesize that this is because the student might be overfitting to its own worse explanations, leading to bad predictions.

In summary, we show stronger evidence for all claims by only using a stronger teacher and we think that a teacher like GPT-4 should yield even better results.

> **Rebuttal on the Motivation**

Some reviewers were also curious about work's motivation. We want to highlight that our objective is not to improve the state-of-the-art accuracy on a certain task. Instead, our goal is to **evaluate and improve** the ability of LLMs to teach via explanations, and our results have positive implications for (1) AIs teaching human students to do things, (2) knowledge distillation into smaller student models, and (3) some form of ToM ability in LMs.

* As for AI teaching human students, consider a vision of LLM-based personal tutors. We want a personalized tutor that teaches students most efficiently. Efficient teaching requires understanding the student behavior and building a mental model of the student (Theory of Mind). Hence, being able to intelligently choose examples to teach (RQ2) will improve teaching efficiency (by saving time and effort) and generating personalized explanations (RQ3) will improve student performance. LLM inference speed is not a significant bottleneck in these situations but communicating explanations is, since teaching each example is expensive.
* From a knowledge distillation perspective, smaller models are easier to deploy in under-resourced scenarios and unlike past work, we address the valuable questions of when and how to distill knowledge from larger to smaller models.
* Finally, our work has implications in the context of AI safety where other agents adopt LLM explanations with unwarranted trust in their correctness. Let’s consider the potential downside of having a good intervention function (RQ2). If a misaligned teacher wants to intentionally harm the students, it can potentially do so by intervening on samples where the student is already good at and **does not** require assistance with. Similarly, if LLMs have the capability to personalize their explanations (RQ3), they can also generate deceiving explanations that purposefully harm the students.

---

### Decision · Program_Chairs · 2023-09-21

**Decision:**

Accept (poster)

**Comment:**

This paper investigates the potential of a Language Model (LLM) as a teacher for other, weaker LLMs by intervening at test time to improve the student LLM's performance. The authors focus on four main research questions (RQs): the effectiveness of teacher intervention, efficient intervention, personalization of explanations, and generalization of teacher explanations. The experiments are conducted using three tasks (StrategyQA, GSM8k, and CommonsenseQA) and four publicly available LLMs (Flan-T5-Large, Flan-T5-XL, Llama-7B, and Llama-13B). The study finds positive evidence for all research questions, but with varying strengths.

The paper proposes an interesting research direction and the experiments are generally well-designed and comprehensive. There are several valid concerns raised by the reviewers but the authors did a very good job addressing them during the discussion period.
I agree with the reviewers that the  language used to describe the connection to Theory of Mind is not adequately supported by the paper. I urge the authors to take this and other comments strongly into consideration for improving the paper.